# A specific prelimbic-nucleus accumbens pathway controls resilience versus vulnerability to food addiction

Laura Domingo-Rodriguez [1,12], Inigo Ruiz de Azua[2,3,12], Eduardo Dominguez[4], Eric Senabre [1], Irene Serra[5], Sami Kummer[1], Mohit Navandar[6], Sarah Baddenhausen [2], Clementine Hofmann[2,7], Raul Andero[8,9,10], Susanne Gerber[6], Marta Navarrete [5], Mara Dierssen [4,11], Beat Lutz[2,3,13], Elena Martín-García [1,8,13] & Rafael Maldonado [1,11,13]*

Food addiction is linked to obesity and eating disorders and is characterized by a loss of behavioral control and compulsive food intake. Here, using a food addiction mouse model, we report that the lack of cannabinoid type-1 receptor in dorsal telencephalic glutamatergic neurons prevents the development of food addiction-like behavior, which is associated with enhanced synaptic excitatory transmission in the medial prefrontal cortex (mPFC) and in the nucleus accumbens (NAc). In contrast, chemogenetic inhibition of neuronal activity in the mPFC-NAc pathway induces compulsive food seeking. Transcriptomic analysis and genetic manipulation identified that increased dopamine D2 receptor expression in the mPFC-NAc pathway promotes the addiction-like phenotype. Our study unravels a new neurobiological mechanism underlying resilience and vulnerability to the development of food addiction, which could pave the way towards novel and efficient interventions for this disorder.

[1] Laboratory of Neuropharmacology-Neurophar, Department of Experimental and Health Sciences, Universitat Pompeu Fabra (UPF), Barcelona, Spain. [2] Institute of Physiological Chemistry, University Medical Center of the Johannes Gutenberg University Mainz, Mainz, Germany. [3] Leibniz Institute for Resilience Research, Mainz, Germany. [4] Centre for genomic regulation (CRG), The Barcelona Institute of Science and Technology, Barcelona, Spain. [5] Instituto Cajal, CSIC, Madrid, Spain. [6] Faculty of Biology and Center of Computational Sciences, Johannes Gutenberg University, Mainz, Germany. [7] Focus Program Translational Neuroscience, Johannes Gutenberg University Mainz, Mainz, Germany. [8] Department of Psychobiology and Methodology in Health Sciences, Universitat Autònoma de Barcelona (UAB), Bellaterra, Spain. [9] Instituto de Salud Carlos III, Centro de Investigació Biomédica en Red de Salud Mental, CIBERSAM, Bellaterra, Spain. [10] Unitat de Neurociència Traslacional, ParcTaulí Hospital Universitari, Institut d'Investigació i Innovació ParcTaulí (I3PT), Institut de Neurociències, UAB, Bellaterra, Spain. [11] Hospital del Mar Medical Research Institute (IMIM), Barcelona, Spain. [12] These authors contributed equally: Laura Domingo-Rodriguez, Inigo Ruiz de Azua. [13] These authors jointly supervised this work: Beat Lutz, Elena Martín-García, Rafael Maldonado. *email: rafael.maldonado@upf.edu

F ood addiction and compulsive eating have recently gained attention due to the increasing worldwide prevalence, the high socio-economical costs, and the lack of effective and long-lasting treatments[1]. This addictive disorder has been closely linked to obesity and eating disorders[1]. The concept of food addiction is still controversial being also under debate if it could represent a behavioral addiction or whether specific components of food could have intrinsic addictive properties similar to drugs of abuse[2]. However, the diagnosis of food addiction is not included in the 5th edition of the Diagnostic and Statistical Manual of Mental Disorders (DSM-5), although a validated tool, the Yale Food Addiction Scale (YFAS) is widely accepted among the scientific community[3].

Food addiction shares common neurobiological mechanisms with drug addiction[4]. Food and drug addiction are complex multifactorial chronic brain disorders that result from the interaction of multiple genes and environmental factors, leading to interindividual variability in the development of the addictive process. Nevertheless, the precise neurobiological mechanisms underlying vulnerability or resilience to food addiction have remained elusive. In order to identify mechanisms at circuit and molecular level, we employed a recently validated animal model that mimics behavioral abnormalities associated with food addiction in humans, such as loss of control, motivation, compulsive eating and impulsivity[5].

Both natural rewards and drugs of abuse exert initial reinforcing effects, targeting the brain reward circuit mainly triggered by dopamine release in the nucleus accumbens (NAc). However, addiction is a result of the transition from initial hedonic intake to loss of control after repeated exposure to the reinforcer, which requires long-lasting adaptations in the reward system and associated pathways. Three major interconnected networks are involved in the development of drug addiction, the basal ganglia, the extended amygdala and the medial prefrontal cortex (mPFC)[6]. The mPFC is involved in the top-down regulation of cognitive flexibility, decision-making and inhibitory control and seems to play a crucial role in the transition to and persistence of the addictive behavior[7–9]. Indeed, there are evidences of a hypoactivity of the mPFC in human addicts[10] and in rats with a compulsive cocaine self-administration[11]. Similarly, positron emission tomography studies described a negative correlation between metabolic activity in prefrontal regions and body weight[12]. In the mPFC, the prelimbic (PL) and the infralimbic (IL) cortex send dense projections to cortical and subcortical regions, including the NAc, which plays a crucial role in behavioral inhibitory control[13]. The mPFC contains excitatory glutamatergic pyramidal neurons and inhibitory GABAergic interneurons, establishing a local network that is modulated in part by the endocannabinoid and the dopaminergic signaling systems[14,15].

The aim of this study was to decipher the neurobiological mechanisms underlying the resilience and the vulnerability to develop food addiction-like behavior. For this purpose, we used innovative tools consisting in a mouse operant behavioral model of food addiction with face validity combined with conditional glutamatergic CB1R mutant mice (Glu-CB1-KO), electrophysiological ex vivo recordings, genome-wide RNA sequencing, chemogenetic interference and adenoviral gene delivery in order to characterize the endophenotype of resilience and vulnerability at a genetic, cellular, circuit and behavioral level. We found that Glu-CB1-KO mice displayed a resilient phenotype to develop food addiction-like behavior, which was associated with an increased excitatory synaptic transmission of glutamatergic neurons in PL and in NAc. In reverse, the silencing of PL neurons projecting to NAc core using a chemogenetic approach induced a phenotype of susceptibility to develop food addiction-like behavior, in particular, a compulsive eating behavior for palatable food. Importantly, transcriptomic analysis of resilient and susceptible mice revealed an upregulation of the dopamine D2 receptor gene (Drd2) in mPFC of food addicted mice. Accordingly, virally-mediated overexpression of dopamine D2 receptor (D2R) specifically in the PL-NAc core projection promoted compulsive eating behavior. Overall, we reveal that the endocannabinoid and the dopaminergic signaling targeting a specific mPFC-NAc pathway control resilience and vulnerability to develop food addiction-like behavior in mice.

## Results

**Glu-CB1-KO mice display resilience to food addiction.** Given that CB1R plays a crucial role in the modulation of glutamatergic transmission[16], we explored the function of this receptor in food addiction-like behavior. Glu-CB1-KO mice[16] lacking CB1R in dorsal telencephalic glutamatergic neurons and their WT littermates were exposed to the recently established operant model of food addiction[5]. Glu-CB1-KO ($n = 58$) and WT mice ($n = 56$) were trained under a fixed ratio (FR) 1 schedule of reinforcement during six sessions followed by 112 sessions under FR5 to obtain chocolate-flavored pellets as reinforcers (Fig. 1a and Methods). In FR1, both genotypes increased the number of reinforcers across sessions without significant differences indicating similar levels of acquisition of the operant conditioning learning. However, when the effort to obtain one single pellet was increased to FR5, the progressive increase of the number of reinforcers was significantly reduced in Glu-CB1-KO as compared to WT (repeated measures ANOVA, genotype effect, $P < 0.001$; interaction genotype x sessions, $P < 0.001$, Fig. 1b), leading to a reduced number of reinforcers over the entire FR5 period. This result suggests that palatable pellets were less reinforcing for the mutants since the FR5 early period ($U$ Mann–Whitney, $P < 0.01$, Supplementary Fig. 1a) and this trait may represent an initial protective factor. Animals that responded <25% of all FR5 sessions and did not achieve the acquisition criteria were excluded from the remaining experimental sequence (14.7% of Glu-CB1-KO and 6.7% of WT).

In the early period (1–18 sessions), a significant genotype difference was only observed in the criterion of compulsivity, evaluated by the number of active responses associated with a footshock delivery. Indeed, a suppressed response in front of negative consequences was revealed in Glu-CB1-KO mice compared to WT mice ($U$ Mann–Whitney, $P < 0.05$, Supplementary Fig. 1b–d). In contrast, non-significant genotype differences were found in the persistence to response, as evaluated by the number of non-reinforced active responses during the pellet-free period, and in motivation, defined by the breaking point obtained during the progressive ratio (PR) schedule (Supplementary Fig. 1b–d). In the late period (95–112 sessions), Glu-CB1-KO mice showed significantly less persistence to response, reduced motivation, and decreased compulsivity in seeking for highly palatable food compared to WT mice, revealing a resilient phenotype of these mutants to develop food addiction ($U$ Mann–Whitney, $P < 0.01$, Fig. 1c–e). Two additional phenotypic traits considered as factors of vulnerability to addiction, impulsivity and sensitive to aversive associative learning, were also evaluated (Supplementary notes, Supplementary Fig. 1e, f). Using the results of three food addiction-like criteria in the late period, we individually categorized mice in addicted (covering 2–3 criteria) or non-addicted, as previously reported[5]. A mouse was considered positive for an addiction-like criterion when its score for each behavior was equal or beyond the 75th percentile of the distribution of the WT group. Only 6.9% of the Glu-CB1-KO mice achieved 2–3 criteria and were considered addicted compared with the 25.0% of WT mice (chi-square, $P < 0.01$, Fig. 1f). Of note, a small percentage of Glu-CB1-KO mice were

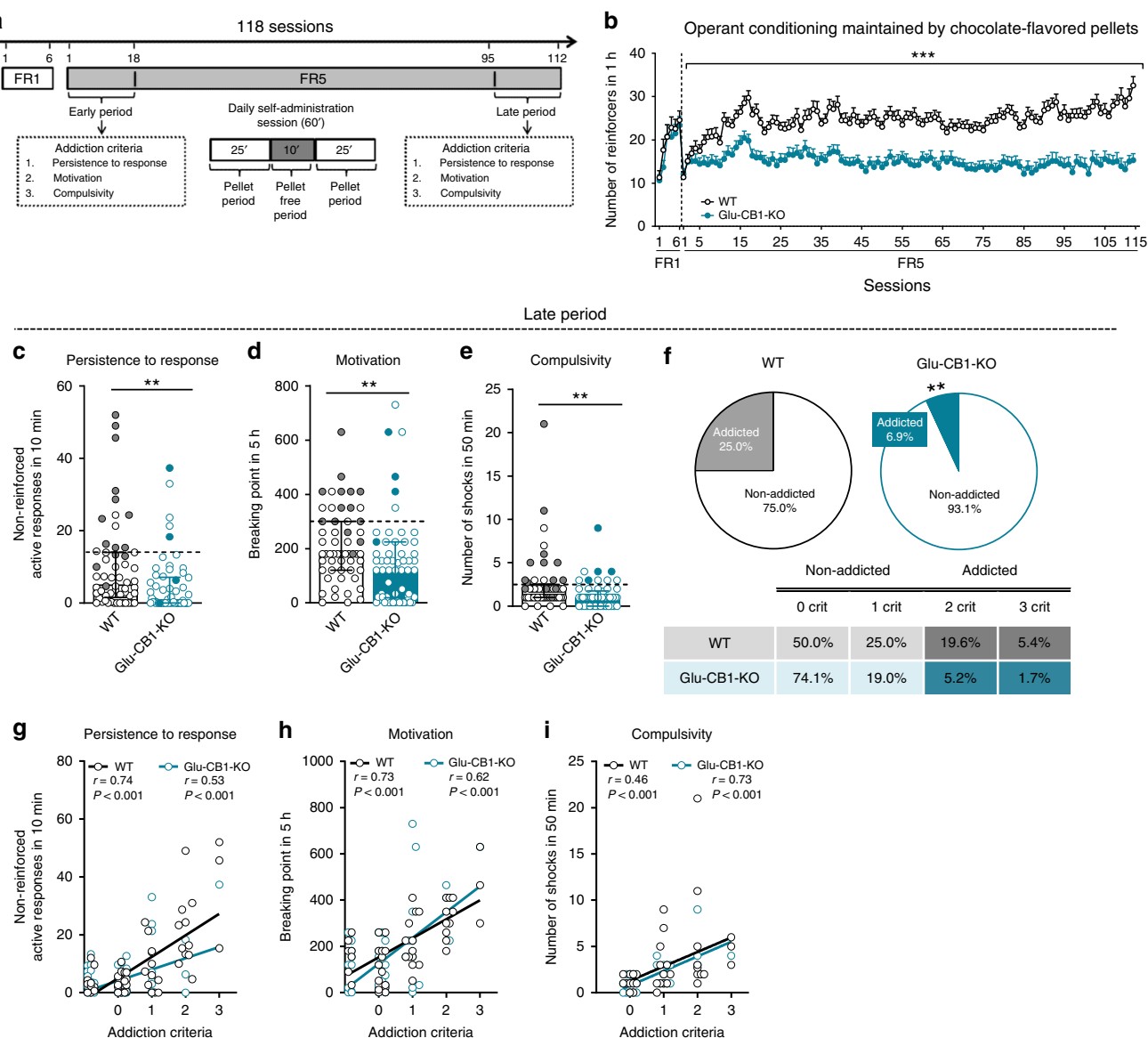

**Fig. 1 Glu-CB1-KO mice display resilience to food addiction. a** Timeline of the experimental sequence of the food addiction mouse model. **b** Reduced number of reinforcers during 1 h of operant training sessions maintained by chocolate-flavored pellets in Glu-CB1-KO compared to WT mice (mean ± S.E.M, repeated measures ANOVA, genotype effect ***$P < 0.001$). **c-e** Glu-CB1-KO mice decreased response in the three addiction-like criteria tests (individual data with median and interquartile range, $U$ Mann–Whitney, **$P < 0.01$): **c** Persistence to response. **d** Motivation. **e** Compulsivity. The dashed horizontal line indicated the 75th percentile of distribution of WT mice, it is used as the threshold to consider a mouse positive for one criterion. Addicted mice in gray filled circles for WT and blue for Glu-CB1-KO mice. **f** Reduced percentage of mice categorized as addicted in Glu-CB1-KO compared to WT mice at the late period (chi-square, **$P < 0.01$). **g-i** Pearson correlations between individual values of addiction-like criteria and **g** non-reinforced active responses in 10 min, **h** breaking point in 5 h and **i** number of shocks in 50 min ($n = 56$ for WT mice and $n = 58$ for Glu-CB1-KO mice; see also Supplementary Figs. 1 and 2; statistical details are included in Supplementary Table 1).

able to become addicted, but the loss of this gene strongly decreased the likelihood to develop food addiction. Positive correlations were found between the number of criteria in both mutant and WT mice and the intensity of each criterion (Fig. 1g–i), and classified addicted mice showed the highest values (Supplementary Fig. 1g–k). Thus, the deletion of the CB₁R in dorsal telencephalic glutamatergic neurons could represent a protective factor to food addiction-like behavior but, as expected in a multifactorial disease, the mutation of one single gene is not enough to totally stop the addiction process.

Previous studies reported that the deletion of CB₁R from dorsal telencephalic glutamatergic neurons has a marked impact on food intake in fasting conditions[17]. However, these mice did not show

body weight changes when were fed ad libitum[17]. Our results show decreased palatable pellets intake by Glu-CB1-KO mice without food restriction ($U$ Mann–Whitney, $P < 0.001$, Fig. 1b). To study if the resilient phenotype of Glu-CB1-KO mice was influenced by the body weight variable, we measured the evolution of the body weight during the whole experimental sequence of 24 weeks, the total average of body weight separately in the early and in late period and the correlations between the body weight and the three addiction-like criteria depending on the genotype (Supplementary Fig. 2a–e). The lack of significant differences between genotypes in the body weight in the early period (Supplementary Fig. 2b), when mutants showed significant increased compulsivity, suggested that the body weight is not a

predisposing factor in the development of the addictive-like behavior. In the late period, body weight differences between genotypes emerged, although the correlations between body weight and the three addiction-like criteria (Supplementary Fig. 2c–e) were not significant indicating that the body weight variable does not explain the lower persistence to response, motivation and compulsivity found in the Glu-CB1-KO mice. For statistical details see Supplementary Table 1.

**Synaptic excitatory transmission is increased in Glu-CB1-KO.** Considering that the activation of presynaptic $CB_1R$ on gluta-matergic neurons is supposed to suppress vesicular release of glutamate[18], and given the notion that increased neuronal activity of the mPFC may regulate the development of addiction[19], we applied ex vivo electrophysiological experiments to uncover the consequences of $CB_1R$ deletion in glutamatergic transmission in mPFC. In naive mutant mice, we quantified the frequency of miniature excitatory postsynaptic currents (mEPSCs) in layer 5 (L5) pyramidal neurons of the PL cortex that project to sub-cortical regions by performing whole-cell recordings in brain slices in the presence of $2 \mu M$ tetrodotoxin (TTX). The frequency of mEPSCs is a readout of the number of neurotransmitter vesicle released[13]. We found that mEPSCs frequency was increased in Glu-CB1-KO compared to WT mice, suggesting an enhanced probability of glutamate vesicle release onto L5 pyramidal PL neurons that could be independent of presynaptic voltage $Ca^{2+}$ ion channels (t-test, $P < 0.01$, Fig. 2a–d). Notably, no significant differences were found in the miniature inhibitory postsynaptic currents (mIPSCs) frequency and in amplitude between Glu-CB1-KO and WT mice (Fig. 2e–g). In a next step, we evaluated if the glutamatergic and GABAergic synaptic transmission in NAc was also altered by the loss of $CB_1R$ in cortical glutamatergic cells. Electrophysiological recordings in the NAc showed an increased mEPSCs frequency and no differences in the mIPSCs in Glu-CB1-KO mice compared to WT mice (t-test, $P < 0.05$, Fig. 2h–n). Thus, the lack of $CB_1R$ in dorsal telencephalic glutamatergic neurons increased synaptic excitatory transmission in PL cortex and NAc without affecting inhibitory synaptic transmission in these areas.

Then, we studied the synaptic facilitation in order to determine whether $Ca^{2+}$-dependent synaptic transmission was affected in Glu-CB1-KO mice. We applied a paired-pulse facilitation (PPF) protocol with a 50 ms interstimulus interval in mPFC L2/3 glutamatergic axons and recorded evoked local field postsynaptic potentials (fPSPs) in L5. We found higher synaptic facilitation (increased PPF ratio) in Glu-CB1-KO than in WT mice, indicating that the lack of $CB_1R$ in glutamatergic terminals also produced an augmented $Ca^{2+}$-dependent synaptic transmission, suggesting an augmented glutamate release (U Mann–Whitney, $P < 0.01$, Fig. 2o–p).

Additionally, we validated the functional deletion of $CB_1R$ signaling at glutamatergic presynaptic terminals in the mPFC of Glu-CB1-KO mice by studying the modulation of the amplitude of single L5 fPSPs evoked by electrical stimulation on L2/3 in the presence of the $CB_1R$ agonist WIN55,212-2 ($5 \mu M$). As expected, the fPSPs amplitude evoked was strongly reduced in the WT mice after WIN55,212-2 application compared to the baseline before treatment, indicating a functional activation of $CB_1R$ signaling (paired t-test, $P < 0.001$, Fig. 2q, r). In contrast, this effect was blunted in conditional knockout mice. As a further demonstration, we completely blocked WIN55,212-2 inhibitory effect in mPFC synaptic transmission by applying the selective $CB_1R$ antagonist rimonabant ($4 \mu M$; Supplementary Fig. 3a, b). Similar results were obtained evaluating the changes of EPSCs amplitude in the NAc. The EPSCs amplitude was strongly reduced in the WT mice after WIN55,212-2 application compared to the baseline.

In Glu-CB1-KO mice, EPSCs amplitude was still slightly diminished by the WIN55,212-2 application, but in a significantly lower extension than in WT mice (paired t-test, $P < 0.01$, Fig. 2s, t). This partial effect of WIN55,212-2 in the mutants could be explained by the expression of $CB_1R$ in glutamatergic projection neurons to the NAc[20–22]. Overall, we obtained a functional confirmation of the absence of the $CB_1R$ in the Glu-CB1-KO mice and demonstrated the modulatory effect of $CB_1R$ in excitatory glutamatergic transmission in PL and NAc.

**Inhibition of PL-NAc core pathway leads to compulsivity.** Based on the increased excitatory transmission in mPFC of resilient Glu-CB1-KO mice, we hypothesized that hypoactivity of glutamatergic transmission in mPFC would promote addictive-like behavior in WT mice when exposed to the palatable food addiction model. To this end, we used a chemogenetic approach to selectively reduce the activity of all the glutamatergic neurons in the PL. Our results revealed that the decreased excitatory transmission in PL promotes food addiction-like behavior (Supplementary notes, Supplementary Fig. 4a–m and Supplementary Fig. 5a–j). Specifically, 42.8% of mice with the inhibition of glutamatergic PL neurons accomplished the criteria of addiction as compared to 15.4% of saline-treated mice (chi-square, $P < 0.01$, Supplementary Fig. 4j), suggesting that a decreased excitability of glutamatergic transmission in PL neurons is involved in the development of this addictive behavior towards highly palatable food. However, PL neurons send projections widely to multiple brain areas, and thus, we asked which specific projection is involved in the loss of behavioral control. To answer this question, we adopted a combined chemogenetic and a retrograde AAV approach[23] that enables the retrograde tagging of neuronal projections. To this end, we injected two AAVs: AAV-hM4Di-DREADD (AAV8-hSyn-DIO-hM4D(Gi)-mCherry) into PL and AAV-retrograde-Cre (AAVrg-pmSyn-EBFP-Cre) into the NAc core (Fig. 3a). Thus, hM4Di receptors expression only occurred in PL neurons that directly projected to NAc core. mCherry and Cre recombinase were visualized by immunofluorescence to verify the injection site of the AAVs and the retrograde transport of Cre (Fig. 3b). Whole-cell current-clamp recordings performed in PL L5 of visually identified hM4Di-mCherry expressing neurons confirmed that clozapine-n-oxide (CNO) activation of hM4Di receptor inhibited the activity of PL-NAc core pyramidal neurons. Indeed, CNO bath application decreased membrane resistance (paired t-test, $P < 0.01$, Supplementary Fig. 6a, b) and subsequently blocked current-evoked action potential firing frequency (paired t-test, $P < 0.05$, Fig. 3c, d). Furthermore, the current to evoke one single action potential in the presence of CNO was higher as compared to the baseline (paired t-test, $P < 0.01$, Supplementary Fig. 6c, d). This effect was hM4Di receptor-specific because no significant differences in the firing rate, membrane resistance nor in rheobase were found when CNO was applied in mPFC slices of mice not expressing the hM4Di receptors (Fig. 3d and Supplementary Fig. 6a–d). Importantly, this PL-NAc core projections inhibited the medium spiny neurons (MSNs) of the NAc as shown by the mEPSCs recordings, the cumulative probability of the amplitude of all signals registered and by the decreased changes of mEPSCs frequency in the NAc after CNO application in mice expressing hM4Di receptors (Fig. 3e, f). No changes in the amplitude nor in the resting membrane potential were reported (Supplementary Fig. 6e–g). Altogether, these results suggested that the chemogenetic inhibition of the PL-NAc core neurons leads to a reduced glutamatergic tone in the NAc.

Using this approach, we expected that selective inhibition of the PL-NAc core projections would decrease the inhibitory control during food operant training, leading to susceptibility to

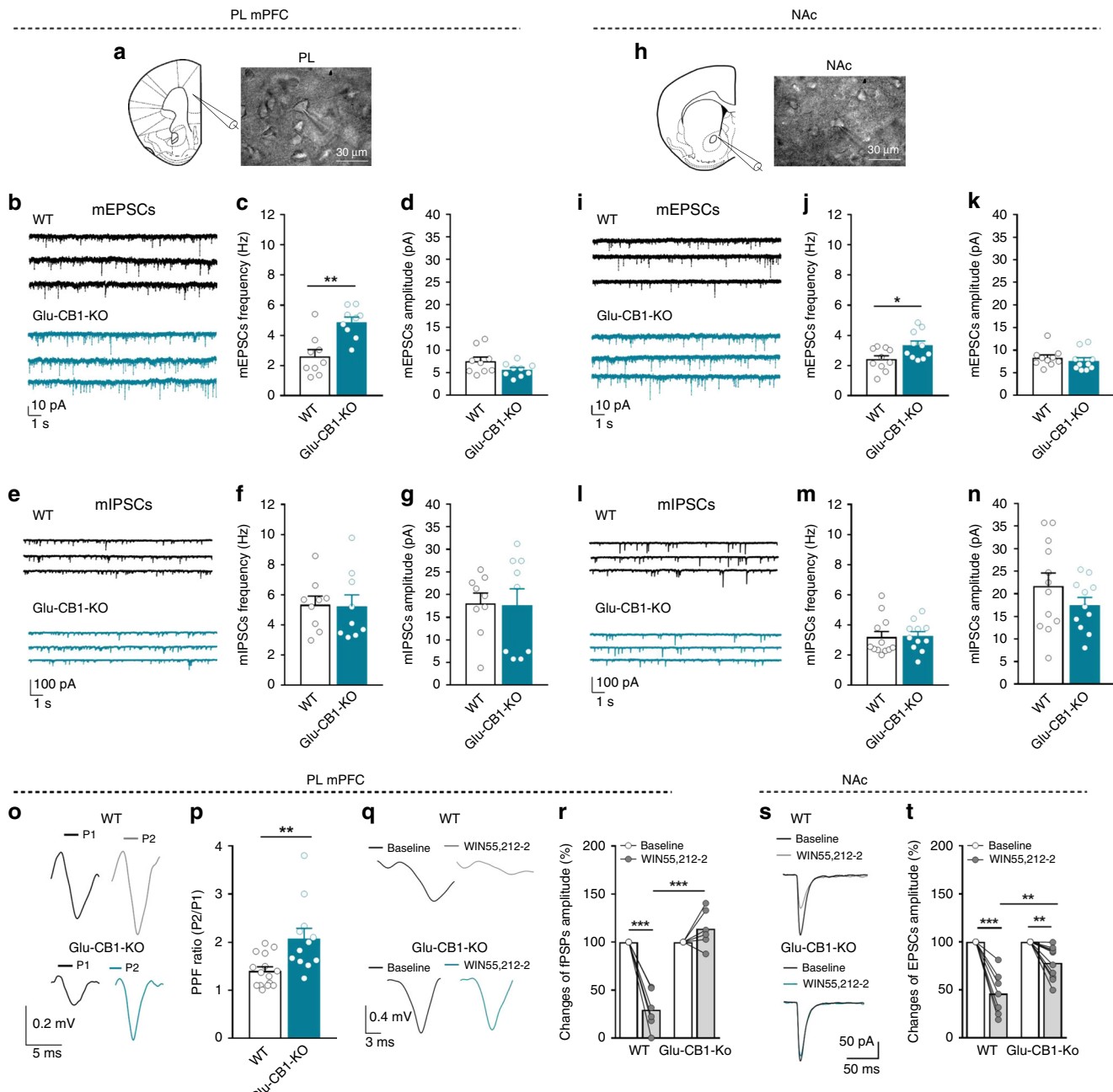

**Fig. 2 Synaptic excitatory transmission is increased in Glu-CB1-KO. a, h** Schematic drawings and infrared differential interference contrast images showing the recorded neuron in the **a** PL and **h** NAc. Scale bars, 30 µm. **b, e** Representative traces of miniature excitatory and inhibitory postsynaptic currents (mEPSCs, mIPSCs) recorded at a holding potential of −70 mV in PL and **i, l** −75 mV in NAc from WT (black traces) and Glu-CB1-KO (blue traces) mice. **c, d** mEPSCs frequency and amplitude and **f, g** mIPSCs frequency and amplitude of WT and Glu-CB1-KO mice in the PL and **j, k, m, n** in the NAc (mean ± S.E.M; t-test, *P < 0.05, **P < 0.01, n = 9–12 cells from n = 12 mice). **o–p** Paired pulse facilitation (PPF). **o** Representative recordings of L5 field postsynaptic potentials (fPSPs) before (P1) and after (P2) stimulating twice in layer 2/3 with an interpulse interval of 50 ms for WT (above) and Glu-CB1-KO (below). **p** Increased paired-pulse facilitation ratio (P2/P1) in Glu-CB1-KO compared to WT (mean ± S.E.M, U Mann–Whitney, **P < 0.01; n = 12–14 slices from n = 5 animals per genotype). **q** Representative recordings of the modulation of fPSPs amplitude before and after application of the CB$_1$R agonist WIN55,212-2 (5 µM) compared to baseline in PL for WT (above) and Glu-CB1-KO (below). **r** Changes of fPSPs amplitude in percentage in PL for WT and Glu-CB1-KO (mean and individual values, paired t-test, ***P < 0.001; n = 6 slices from n = 3 animals per genotype). **s** Representative synaptic responses showing mean EPSCs (five consecutive EPSCs) before and after CB$_1$R agonist WIN55,212-2 (5 µM) bath application in the NAc for WT (above) and Glu-CB1-KO (below). **t** Average relative changes of EPSCs amplitude in NAc of WT and Glu-CB1-KO mice (mean and individual values, paired t-test, **P < 0.01, ***P < 0.001; n = 7–10 cells from n = 4 animals per genotype; see also Supplementary Fig. 3; statistical details are included in Supplementary Table 2).

develop addiction-like behavior. Therefore, we used the early food addiction protocol (Fig. 3g) to train WT mice (n = 32) to self-administer chocolate-flavored pellets in the operant chambers under FR1 (two sessions) and FR5 (three sessions) schedule of

reinforcement before AAVs injection and under FR5 (four sessions) after injection to recover the basal levels of responding. Then, an osmotic minipump filled with CNO (n = 22) or saline (n = 12) was subcutaneously implanted in the back of each

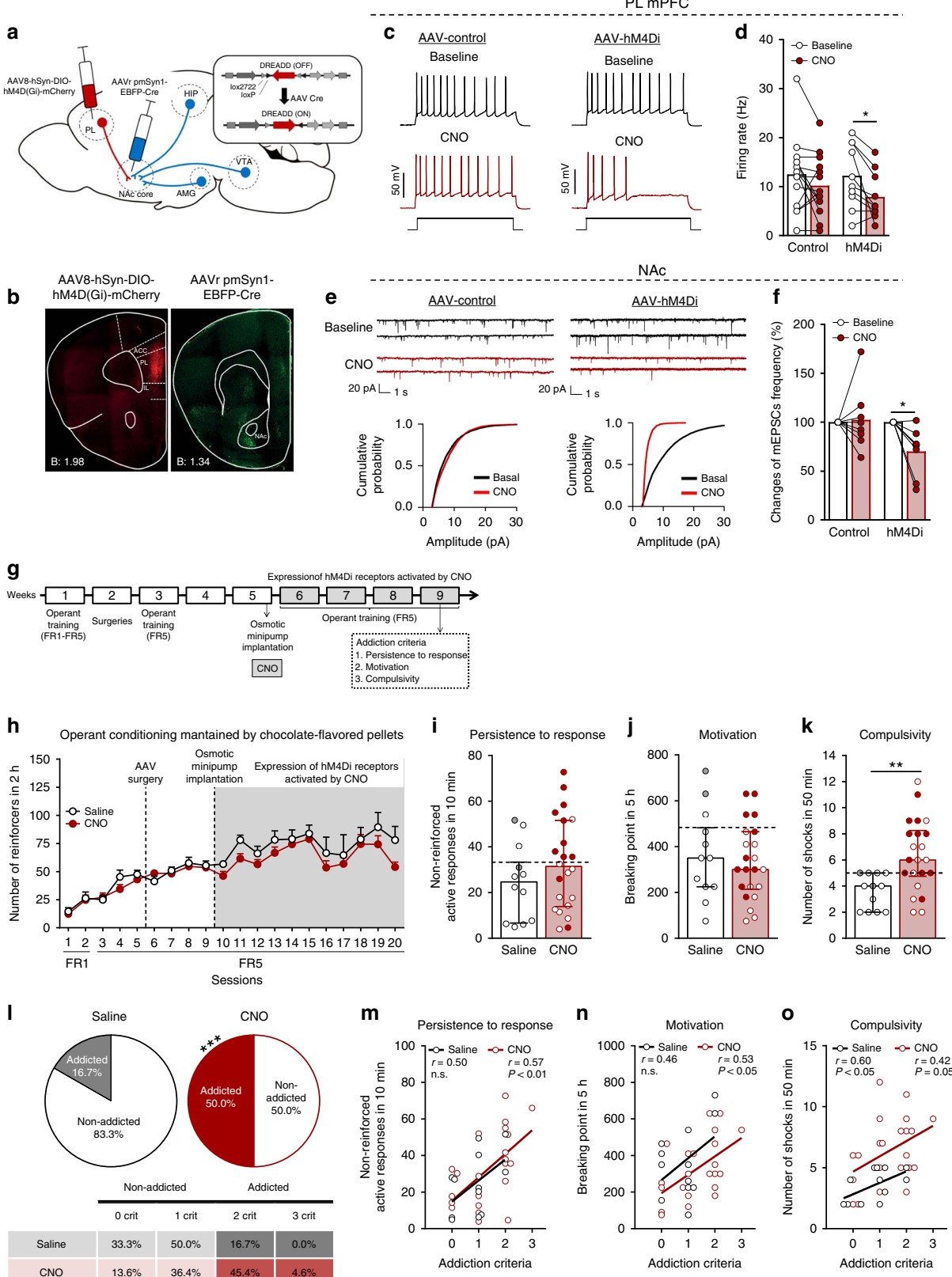

mouse. During the chronic CNO exposure (four weeks, 0.25 μl/h) with the subsequent inhibition of the glutamatergic PL-NAc core neurons, mice underwent FR5 sessions for four weeks, and the three food addiction-like criteria were evaluated during the last week. No differences were found in the number of reinforcers in the daily training sessions between CNO and saline-treated mice

during the operant conditioning maintained by chocolate-flavored pellets, indicating that the inhibition of PL-NAc core projection did not affect the reinforcing effects of these pellets (Fig. 3h). In contrast, this specific manipulation of the PL-NAc projection produced a robust increase in the food addiction compulsivity criterion (t-test, P < 0.01, Fig. 3i–k). Thus, mice with

**Fig. 3 Inhibition of PL-NAc core pathway leads to compulsivity. a** Scheme of combinatorial viral strategy for selective hM4Di-mCherry expression in PL-NAc core neurons. **b** Representative immunofluorescence images of Cre-dependent hM4Di-mCherry detected at PL injection site (left) and Cre recombinase at NAc core (right). **c–f** Chemogenetic inhibition of glutamatergic PL-NAc core neurons induces changes in excitatory transmission in the PL and NAc core. **c** Representative recordings showing evoked (200 pA) action potential in WT mice injected with AAV-control (left) and AAV-hM4Di (right) in PL layer 5 mCherry visualized neurons at baseline and after CNO (10 μM) application. **d** Quantification of the firing rate (Hz) (mean and individual values; paired $t$-test, *$P < 0.05$; $n = 14$ cells from $n = 4$ mice injected with AAV-control and $n = 10$ cells from $n = 4$ animals injected with AAV-hM4Di). **e** Representative traces of miniature excitatory postsynaptic currents (mEPSCs) recorded at a holding potential of $-75$ mV in NAc from mice injected with AAV-control (left) or AAV-hM4Di (right) in baseline and after CNO (10 μM) bath application (above). Cumulative probability plot of the mEPSCs amplitude from both control (left) and hM4Di (right) expressing mice in baseline conditions and after CNO (10 μM) bath application (below). **f** Changes of mEPSCs frequency in percentage (mean and individual values; paired $t$-test, *$P < 0.05$; $n = 8$ cells from $n = 6$ mice injected with AAV-control and $n = 8$ cells from $n = 7$ mice injected with AAV-hM4Di). **g** Timeline of the experimental sequence of the early period of food addiction mouse model. **h** Number of reinforcers during operant training sessions maintained by chocolate-flavored pellets (mean ± S.E.M). **i–k** Behavioral tests of the three addiction-like criteria showing increased compulsivity in CNO-treated mice (individual values with the median and the interquartile range, $U$ Mann–Whitney, **$P < 0.01$). The 75th percentile of distribution of mice treated with saline is indicated by the dashed horizontal line. Addicted mice in gray filled circles for saline-treated mice and red for CNO-treated mice. **l** Increased percentage of CNO-treated mice classified as food addicted animals (chi-square, ***$P < 0.001$). **m–o** Pearson correlations between individual addiction-like criteria and **m** non-reinforced active responses in 10 min, **n** breaking point in 5 h, **o** number of shocks in 50 min ($n = 12$ for saline-treated mice and $n = 22$ for CNO-treated mice; PL prelimbic, NAc nucleus accumbens, Amg amygdala, Hip hippocampus, VTA ventral tegmental area, ACC anterior cingulate cortex, IL infralimbic; see also Supplementary Fig. 6; statistical details are included in Supplementary Table 4).

chronic inhibition of PL-NAc core projection could not stop responding for chocolate-flavored pellets in the shock test, receiving higher number of shocks compared to saline group. Persistence to response and motivation for chocolate-flavored pellets were not significantly different in these animals (Fig. 3i–k). The increased compulsivity could not be explained by an unspecific effect of CNO, since no significant differences of CNO on the three addiction-like criteria were detected in chronically treated mice not expressing the inhibitory DREADD (Supplementary Fig. 5a–d). In addition, no side effects of CNO were observed on body weight, food intake or locomotor activity in these mice (Supplementary Fig. 6h–j). We observed that our manipulation produced a vulnerable phenotype to develop food addiction, as shown by the fact that the majority of CNO-treated animals (77.3%) were above the 75th percentile of the compulsivity criterion. Using the categorization based on the three criteria, a highly significant percentage of the CNO-treated mice (50.0%) was considered addicted as compared to the control animals (16.7%, chi-square, $P < 0.001$, Fig. 3l). Finally, a significant positive correlation between the number of criteria reached and the values of each addiction-like criterion was found in CNO group in the three criteria and in saline group only in the criterion of compulsivity (Fig. 3m–o), and addicted mice showed higher values in both saline and CNO-treated mice (Supplementary Fig. 6k–m). Thus, we demonstrated that the specific inhibition of the PL neurons projecting to the NAc core is a crucial network that confers vulnerability to develop food addiction.

**Drd2 gene expression is upregulated in mPFC of addicted mice.** To characterize gene expression signatures for food addiction, we performed whole transcriptome analysis of mPFC (Supplementary Fig. 7a) of addicted and non-addicted WT and Glu-CB1-KO mice, classified on the bases of the performance at the late period ($n = 4–6$). First, we selected both WT and Glu-CB1-KO mice displaying similar extreme values in the three addiction-like criteria (Fig. 4a–c). In these mice, non-significant differences were observed in pellets intake between addicted and non-addicted mice in the last FR5 session, immediately before tissue collection (Fig. 4d). Therefore, we assume that changes in gene expression would be related to the addiction phenotype and not to the amount of pellets intake during the last training session. To determine overall transcriptional changes in the addicted vs. non-addicted mice, we applied principle component analysis (PCA), revealing the variation between the samples.

Two different clusters were observed for the addicted and non-addicted mice (Supplementary Fig. 7b). Upon performing differential gene expression analysis between non-addicted and addicted mice, 31 genes were significantly upregulated, whereas 70 genes were downregulated (Fig. 4e and Supplementary Tables 7 and 8). Interestingly, genes previously related to the reward system such as *Drd2* (dopamine receptor type 2), *Adora2A* (adenosine receptor 2a), *Gpr88* (orphan G-protein coupled receptor 88), and *Drd1* (dopamine receptor type 1) mRNA were found to be upregulated in the addicted mice. Several genes such as *Myh11* (myosin heavy chain 11), *Acta2* (actin alpha 2), *Cdh1* (cadherin 1), *Ptgds* (prostaglandin D2 synthase), and *Fosb* (fosb proto-oncogene, AP-1 transcription factor subunit) were downregulated, suggesting changes in neuronal plasticity, prostaglandin synthesis and gene regulation[24–27]. The four differentially upregulated genes were selected for technical validation by quantitative PCR (qPCR). The results confirmed the upregulation of *Drd2*, *Adora2A*, *Gpr88*, and *Drd1* mRNAs in addicted mice ($P < 0.05$, Fig. 4f). As expected, no changes were found in housekeeping genes between non-addicted and addicted mice samples (Supplementary Fig. 7c–e), validating RNA-seq data analysis.

Furthermore, RNA-seq data analysis also showed differentially expressed genes between WT and Glu-CB1-KO mice: 18 genes were significantly upregulated, whereas 26 genes were downregulated (Fig. 4g and Supplementary Tables 9 and 10). Technical validation by qPCR confirmed that *Cnr1* (cannabinoid type 1 receptor) was downregulated as expected, and *Fos* (c-fos) was also downregulated in mutants ($t$-test, $P < 0.05$, $U$ Mann–Whitney, $P < 0.001$, Fig. 4h). *Dnah6*, *Spaca1*, and some small nucleolar RNAs (Snora68, 78, 70, 8, 31) were upregulated. Interestingly, c-fos has been extensively used as molecular marker of neuronal activity, implicating an enhanced neuronal activity in mPFC of WT compared to Glu-CB1-KO mice. A non-significant difference was found for *Npas4* mRNA (Fig. 4h), encoding the transcription factor neuronal PAS containing protein 4. This result confirmed the observation that Npas4 was not differentially expressed according to established thresholds of RNA-seq analysis (Fig. 4g).

In summary, transcriptomic data analysis shed new light into the gene expression signature in mPFC related to food addiction, suggesting molecular mechanisms associated with the loss of control over palatable food intake. As we found that *Drd2* gene is the most significantly upregulated gene in addicted mice, we hypothesized that this upregulation could play a key role in the

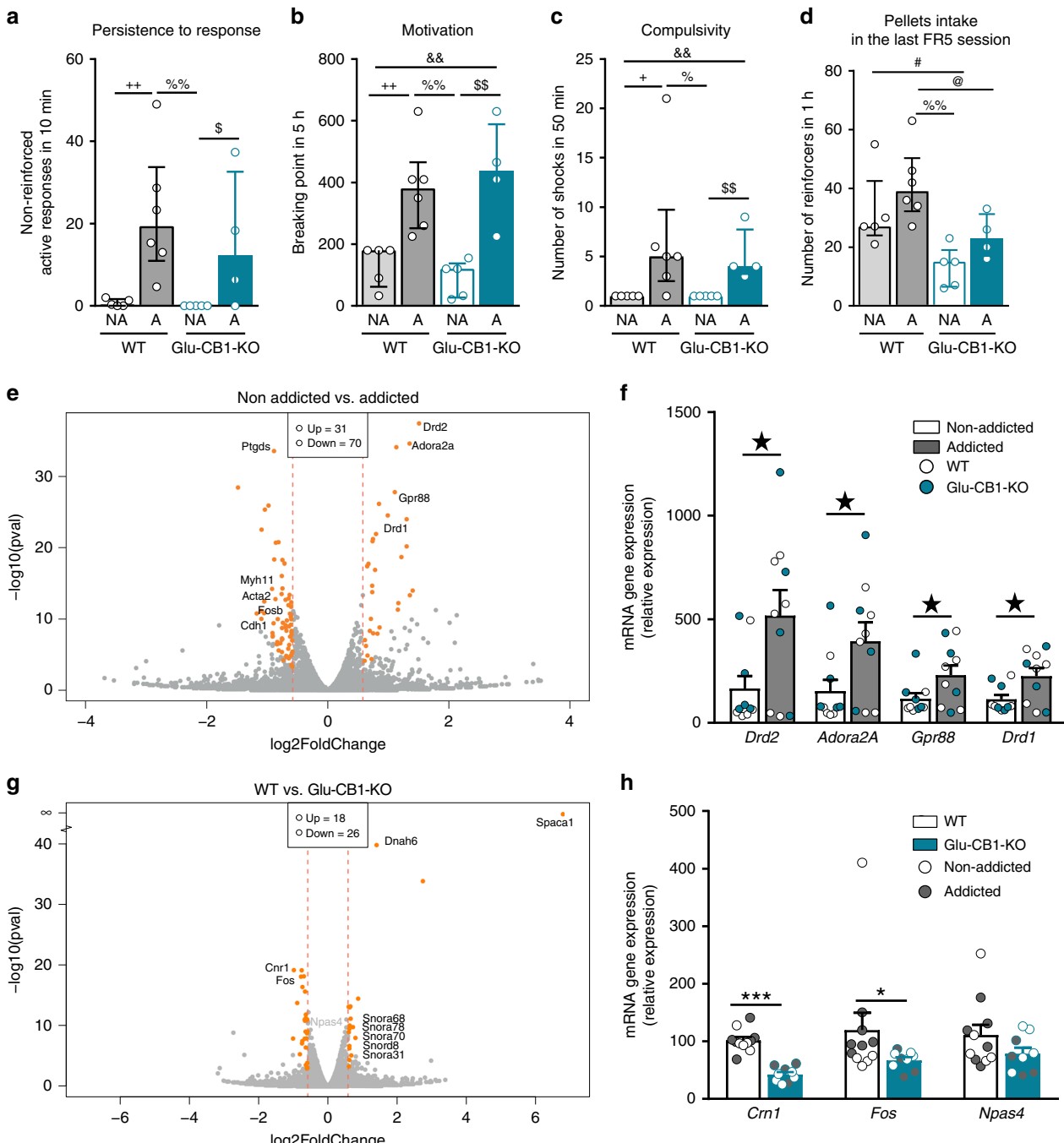

**Fig. 4 *Drd2* gene expression is upregulated in mPFC of addicted mice. a–c** Behavioral tests of the three addiction-like criteria during the late period for those mice selected for RNA-seq in each of the four groups, addicted (A) and non-addicted (NA) mice in both genotypes (individual values and bars with median and the interquartile range; $n = 5$ WT NA, $n = 6$ WT A, $n = 5$ Glu-CB1-KO NA, $n = 4$ Glu-CB1-KO A). **a** Persistence to response. **b** Motivation. **c** Compulsivity. **d** Pellets intake in the last FR5 session before sample collection. **e, g** Volcano plot of the RNA-seq data analysis. The cutoff of 1.5-fold change and differentially expressed genes are highlighted in orange. **e** Differentially expressed genes in addicted mice compared to non-addicted mice. **g** Differentially expressed genes in Glu-CB1-KO compared to WT mice. **f, h** Quantitative real-time PCR of selected genes. ($+ P < 0.05, + + P < 0.01$ WT NA vs. WT A, $\#P < 0.05$ WT NA vs. Glu-CB1-KO NA, $\&\&P < 0.01$ WT NA vs. Glu-CB1-KO A, $\%P < 0.05$, $\%\%P < 0.01$ WT A vs. Glu-CB1-KO NA, $@P < 0.05$, WT A vs. Glu-CB1-KO A, $\$P < 0.05$, $\$\$P < 0.05$ Glu-CB1-KO NA vs. Glu-CB1-KO A; ★$P < 0.05$ non-addicted vs. addicted; $*P < 0.05$, $***P < 0.001$ WT vs. Glu-CB1-KO; see also Supplementary Fig. 7; statistical details are included in Supplementary Table 5).

development of food addiction-like behavior, irrespective of the presence or absence of CB$_1$R.

**_Drd2_ overexpression in PL-NAc core pathway promotes compulsivity.** Based on the above findings, we tested whether the

selective overexpression of *Drd2* in the PL-NAc core projections induces the loss of inhibitory control for palatable food self-administration. Using these experimental conditions, we aimed at mimicking the upregulation of the *Drd2* gene observed in addicted mPFC after long-term exposure to highly palatable food

operant training. First, we confirmed under basal conditions low endogenous *Drd2* mRNA expression in PL as compared to NAc and caudate putamen by in situ hybridization (ISH) (Supplementary Fig. 8a).

For specific overexpression in PL-NAc core projections, we used a dual viral vector approach with an Cre-dependent AAV-$D_2R$ (AAV-hSyn-DIO-D2L-mVenus, $n = 13$) and AAV-control (AAV-Syn1-Stop-GFP, $n = 12$) injected into PL, and an AAV-retrograde-Cre (AAV-pmSyn-EBFP-Cre) injected into the NAc core (Fig. 5a). We first verified the AAV injection site by immunofluorescence against mVenus and Cre recombinase (Fig. 5b). We also confirmed by ISH that retrogradely expressed Cre recombinase mRNA is present in PL and, importantly, approximately 50% of endogenous *Drd2* mRNA-positive cells revealed co-expression with *Cre* mRNA in PL (Supplementary Fig. 8b, c). Quantitative real-time PCR showed 40-fold increased levels of *Drd2* gene expression in mice overexpressing $D_2R$ as compared to control mice in the mPFC (Supplementary Fig. 8d). Additionally, immunohistochemical experiments revealed over-expressed $D_2Rs$ in the neuropil of the PL cortical neurons (Supplementary Fig. 8e–r). The functional consequence of *Drd2* overexpression was first investigated by electrophysiology. We performed in vitro whole-cell recordings in brain slices using the $D_2R$ selective agonist quinpirole to confirm that the over-expression of $D_2R$ decreased the excitability of PL-NAc core projection neurons. Quinpirole ($2 \mu M$) application significantly increased rheobase and reduced membrane resistance and firing rate in response to a 150 pA current square pulse (paired $t$-test, $P < 0.05$, Fig. 5c, d and Wilcoxon test, $P < 0.01$, Supplementary Fig. 9a–d). No differences were observed in control PL L5 pyramidal neurons, suggesting that the $D_2R$ overexpression was responsible of this inhibitory effect despite the fact that $D_2R$ is also endogenously expressed in mPFC of control mice (Fig. 5c, d and Supplementary Fig. 9a–d). The lack of effect after quinpirole application in control animals could be explained by the low levels of endogenous $D_2R$ in mPFC according to RNA-seq data. In agreement, our ISH experiments showed endogenous *Drd2* mRNA expression in this type of cells (L5), but at very low levels close to the limit of detection. Furthermore, we investigated whether this reduced excitability in the PL-NAc core neurons modulates the synaptic glutamatergic transmission in the NAc. Whole-cell recordings in the NAc confirmed a reduction in the changes of mEPSCs frequency accompanied by a sustained difference in the cumulative probability of the amplitude of all the signals registered (Fig. 5e, f). No changes in the amplitude nor in the resting membrane potential were reported (Supplementary Fig. 9e–f). Additionally, dopamine application ($10 \mu M$) in the PL L5 neurons showed a reduction in membrane resistance, firing rate and increased rheobase (Wilcoxon test, $P < 0.05$, $P < 0.01$, paired $t$-test, $P < 0.001$, Supplementary Fig. 9g–l). Therefore, these electrophysiological results reveal that the overexpression of $D_2R$ at glutamatergic PL neurons projecting to the NAc has an effect in both areas, the PL and NAc.

For the behavioral analysis, we used a food addiction procedure during the early period similar to experiments shown above (Fig. 5g). We found that overexpression of $D_2R$ in PL-NAc core projection neurons produced compulsive behavior towards highly palatable food, despite harmful consequences in the shock test. This manipulation only affected the compulsivity addiction criterion, since no differences were found in the persistence to response, motivation nor reinforcement ($t$-test, $P < 0.05$, Fig. 5h–k). A subset of 57.1% of manipulated mice were above the 75th percentile threshold of the control group in the compulsivity criterion, and this result was not found in the other addiction-like criteria. Finally, the percentage of $D_2R$ over-expressing mice that achieved 2–3 addiction-like criteria was

30.8% compared to 8.3% in control mice (chi-square, $P < 0.01$, Fig. 5l). A positive correlation between the number of criteria reached and the values obtained in each criterion was found in all the groups and addicted animals showed extreme values (Fig. 5m–o, Supplementary Fig. 10a–c). No differences were found in additional variables such as body weight, food intake and locomotor activity after $D_2R$ overexpression (Supplementary Fig. 10d–f). In summary, we revealed that overexpression of *Drd2* allowed that dopamine via $D_2Rs$ decreased the excitability of PL-NAc core projections, conveying the vulnerability to develop food addiction.

## Discussion

The easy access to hypercaloric and palatable foods in the western lifestyle is a major contributing factor for developing food addiction and obesity. However, it is still unclear why some individuals get loss of control over food intake and develop food addiction, whereas others are resilient. In this study, we describe a novel neurobiological mechanism underlying the resilience and vulnerability to food addiction-like behavior targeting excitatory glutamatergic transmission in PL-NAc core projection, which appears to be modulated by the endocannabinoid and the dopaminergic signaling systems.

We first studied the phenotype of food addiction in conditional Glu-CB1-KO mice and their control littermates using a behavioral animal model with high-translational face validity to human addiction[5]. Here, we mimicked the transition to addiction after repeated seeking of palatable food in long operant training. Our findings revealed that the lack of $CB_1R$ in dorsal telencephalic glutamatergic neurons induced a strong resilience to food addiction as revealed by the significantly reduced percentage (6.9%) of addicted mice in the mutant group. Glu-CB1-KO mice were characterized by less perseverance, reduced motivation and decreased compulsivity for highly palatable food. Moreover, a phenotype of improved learning in a cocaine self-administration paradigm[28], reduced exploratory behavior[29], increased neophobia, high passive fear response after conditioning, and decreased food intake after fasting was described in these mutants[17,30]. A previous study reported an effect on food intake when $CB_1R$ was deleted from dorsal telencephalic glutamatergic neurons but exclusively after a 24 h fasting period[17]. Our results suggest a decreased reinforcing effects of chocolate-flavored pellets in the mutants, which could contribute to the protective phenotype to develop the multifactorial food addiction disorder. Notably, Glu-CB1-KO mice did not show changes in body weight in the early period similarly to what was previously reported when mice were fed ad libitum with regular chow in a 12 weeks period of time[17]. In the late period, mutants showed lower body weight than WT mice but this variable did not affect the resilient phenotype since no correlations between the three addiction-like criteria and body weight were revealed.

In our study, Glu-CB1-KO mice showed an enhanced inhibitory control to palatable food operant seeking that strongly reduced the transition from controlled to compulsive seeking for highly palatable food. This resilient phenotype was mediated by increased synaptic glutamatergic transmission in mPFC and in NAc. Specifically, our electrophysiological recordings of mEPSCs frequency in Glu-CB1-KO mice revealed an enhanced excitatory synaptic transmission onto pyramidal glutamatergic neurons in L5 of the PL cortex and in their targeted MSNs in the NAc. This effect was specific of glutamatergic transmission, since the synaptic GABAergic transmission in these mutants was not altered in both brain regions. Additionally, the PPF results revealed that the deletion of presynaptic excitatory $CB_1R$

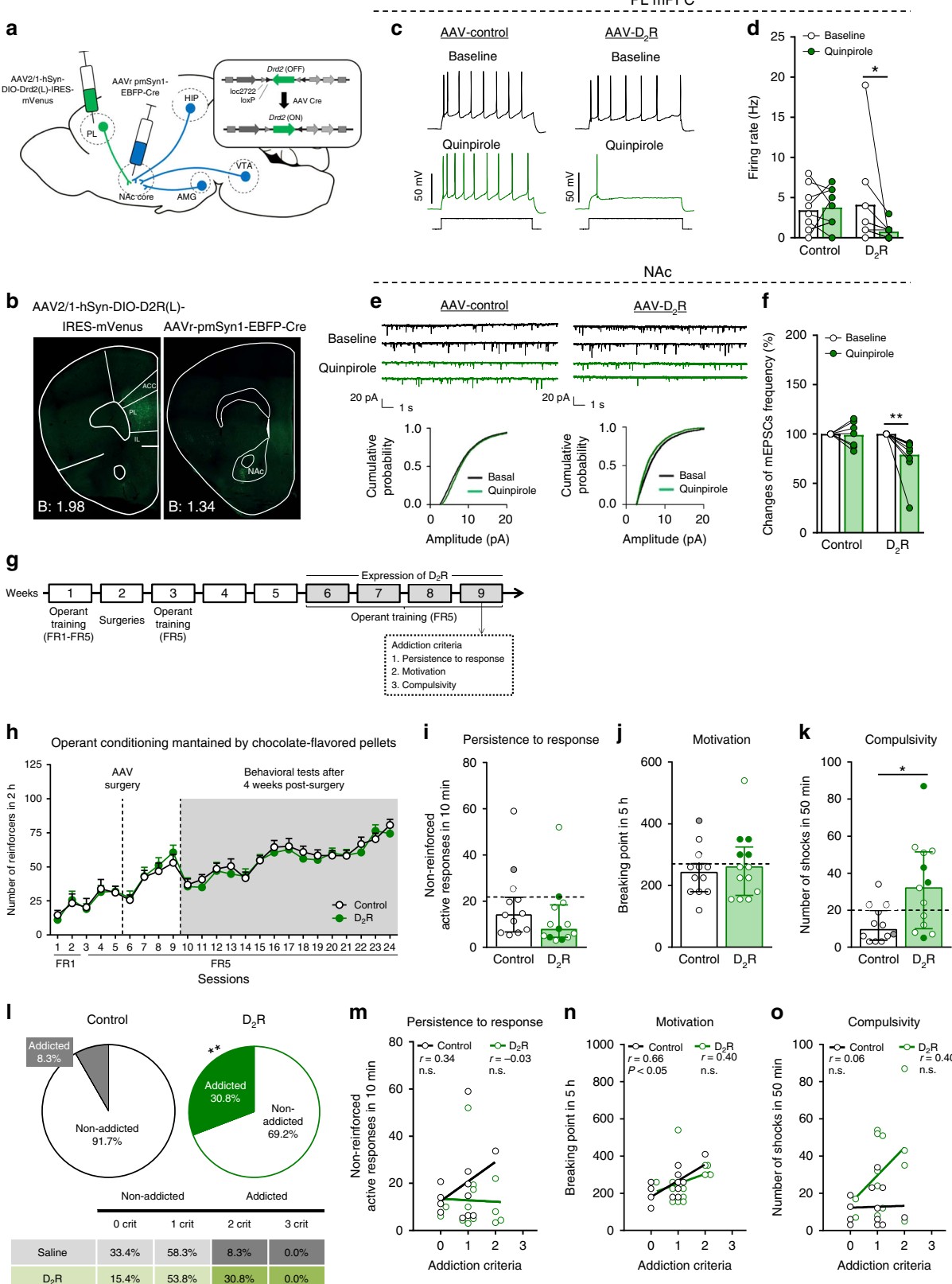

produced an enhancement in glutamate vesicle release in the PL cortex that is action potential dependent. Altogether, these results highlight the involvement of the glutamatergic PL-NAc projections in the behavioral phenotype of inhibitory control observed in Glu-CB1-KO mice. According to this protective phenotype, animal studies targeting the $CB_1R$ by pharmacological blockade or genetic deletion showed decreased addiction-like behavior[5], but the precise pathways and cell types involved have not been characterized. In our study, we revealed that the main cells involved are the dorsal telencephalic glutamatergic neurons that project to the NAc. Although $CB_1R$ on glutamatergic neurons are less abundant than in GABAergic neurons[31], the signal

**Fig. 5 Drd2 overexpression in PL-NAc core pathway promotes compulsivity. a** Scheme of combinatorial viral strategy for selective $D_2R$-mVenus expression in PL-NAc core projecting neurons. **b** Representative immunofluorescence images showing Cre-induced $D_2R$-mVenus protein at PL injection site (left) and Cre recombinase protein at NAc core injection site (right). **c**-**f** Overexpression of $D_2R$ in PL-NAc core neurons induces changes in excitatory transmission in the PL and NAc core. **c** Representative recordings showing evoked (150 pA) action potential in WT mice injected with AAV-control (left) and AAV-$D_2R$ (right) in PL layer 5 mVenus visualized neurons at baseline and after quinpirole (2 μM) application. **d** Quantification of the firing rate (Hz) (mean and individual values; Wilcoxon test, *$P < 0.05$, $n = 9$ cells from $n = 3$ animals injected with AAV-control and $n = 9$ cells from $n = 3$ animals overexpressing $D_2R$). **e** Representative traces of miniature excitatory postsynaptic currents (mEPSCs) recorded at a holding potential of $-75$ mV in NAc from mice injected with AAV-control (left) or AAV-$D_2R$ (right) in baseline and after quinpirole application (above). Cumulative probability plot of the mEPSC amplitudes from both control (left) and $D_2R$ (right) expressing mice in baseline conditions and after quinpirole bath application (below). **f** Changes of mEPSCs frequency in percentage (mean and individual values; Wilcoxon test, **$P < 0.01$, $n = 7$ cells from $n = 5$ animals injected with AAV-control and $n = 10$ cells from $n = 7$ animals overexpressing $D_2R$). **g** Timeline of the experimental sequence of the early period of food addiction mouse model. **h** Number of reinforcers during operant training sessions maintained by chocolate-flavored pellets. **i**-**k** Behavioral tests of the three addiction-like criteria in the early period showed increased compulsivity in mice overexpressing $D_2R$ (individual values with the median and the interquartile range, $t$-test, *$P < 0.05$). The 75th percentile of distribution of control mice is indicated by the dashed horizontal line. Addicted mice in gray filled circles for control and green for $D_2R$ mice. **l** Increased percentage of mice overexpressing $D_2R$ classified as food addicted animals (chi-square, **$P < 0.01$). **m**-**o** Pearson correlations between individual addiction-like criteria and **m** non-reinforced active responses in 10 min, **n** breaking point in 5 h, **o** number of shocks in 50 min ($n = 12$ for control mice and $n = 13$ for $D_2R$ mice; PL prelimbic, NAc nucleus accumbens, Amg amygdala, Hip hippocampus, VTA ventral tegmental area, ACC anterior cingulate cortex, IL infralimbic; see also Supplementary Figs. 8–10; statistical details are included in Supplementary Table 6).

transduction processes appear to be more efficient in glutamatergic than in GABAergic neurons[32].

Furthermore, an additional experiment of electrophysiology ex vivo in the mPFC and NAc with the application of the $CB_1R$ agonist WIN55,212-2 in mutants and in WT mice has been performed. With this experiment, we validated at functional level the deletion of $CB_1R$ at glutamatergic presynaptic terminals both in PL and in NAc of Glu-CB1-KO. As it was expected, the amplitude of fPSPs in the mPFC and the EPSCs in the NAc were strongly reduced in WT mice after WIN55,212-2 application compared to the baseline, which indicates a functional activation of presynaptic $CB_1R$. In the mutants, the fPSPs amplitude was not diminished in the mPFC confirming the absence of the $CB_1R$ at presynaptic glutamatergic terminals. On the other, in the NAc, the EPSCs amplitude was slightly decreased after WIN55,212-2 application, but in a significantly lower extend than in WT mice possibly due to the presence of $CB_1R$ in other cell-types or other glutamatergic projecting neurons to the NAc[20–22]. PL pyramidal neurons projecting to the NAc showed increased excitatory transmission due to the lack of the $CB_1R$ and we therefore postulate that they stimulate the GABAergic D2-MSNs indirect pathway, thereby facilitating the avoidance behavior (NO GO response) (Supplementary Fig. 11a). This top-down mechanism involving PL-NAc projections seems crucial in the resilient phenotype of the mutants and was further investigated in our study. Importantly, addiction-like behavior is associated with an alteration in the synaptic excitatory transmission of prefrontal cortical areas that normally provide inhibitory control striatal-mediated behavior[33].

Previous data pointed out that the PL subregion of the mPFC is particularly involved in drug addiction[34]. Indeed, inhibition of the PL cortex by in vivo optogenetics increased compulsive cocaine intake in rats[11], and the repetitive transcranial magnetic stimulation applied to the human dorsolateral PFC, equivalent to the rodents PL cortex, reduced cocaine use and craving[35]. Therefore, we decreased the glutamatergic transmission of the PL region by using a Cre-dependent inhibitory chemogenetic approach in Nex-Cre mice. CNO-induced silencing of glutamatergic neurons in hM4Di-injected mice increased the percentage of addicted animals. These results revealed a crucial role of this cortical region in the development of food addiction-like behavior. We cannot discard the possible involvement of the anterior cingulate cortex in this behavior, since we also observe residual viral expression in this brain region. This area is anatomically grouped with the PL region to the dorsal mPFC[36], that

corresponds to the dorsolateral PFC in humans[37], an area classically involved in response inhibition[34]. However, it remains still unknown, which is the specific downstream target of PL projections involved in food addiction. Pyramidal glutamatergic neurons of the PL cortex project to different brain areas, such as hippocampus, ventral tegmental area (VTA), amygdala and NAc, among others, conferring to the mPFC a complex connectivity role as a central hub of communications[34]. Considering that the PL area preferentially projects to the core part of the NAc[13], and the functional role of this pathway[38], we specifically targeted the PL-NAc core pathway using a dual viral vector approach. We injected an AAV expressing a Cre-dependent inhibitory DREADD in the PL and a retrograde AAV-variant expressing Cre recombinase into the NAc core. Using this dual viral vector approach in combination with a chronic delivery of CNO ligand, we found that silencing the PL-NAc core projections enhanced specifically the compulsive eating behavior, triggering that animals could not stop palatable food self-administration despite negative consequences. The other addiction-like criteria, persistence to response and motivation, were not modified by chemogenetic interference. These criteria represent different endophenotypes from compulsivity and consequently, the neuronal pathways recruited could be different. Persistence to response reflects the difficulty of mice to stop food seeking. This behavior is related to a persistent desire or unsuccessful efforts to cut down the response due to a habit formation or disruption of extinction learning and has been reported to involve dorsal striatal and hippocampal pathways[39]. In turn, motivation is apparently directly related to reward processing and has been associated with VTA-NAc pathways[4]. The successful silencing of PL-NAc core cortical pyramidal neurons was demonstrated by patch-clamp experiments, showing a decreased firing rate and membrane resistance in neurons expressing hM4Di receptors exclusively in the presence of CNO, confirming that the driving compulsive food seeking was underlined by a reduced PL-NAc core activity. Future studies should be performed to elucidate the role of other pathways from the PL to other downstream areas, such as the amygdala, in the development of food addiction-like behavior.

A small percentage of Glu-CB1-KO mice became addicted after the prolonged highly palatable food exposure. Therefore, the lack of a single gene was not enough to totally block the transition to addiction, as expected for a multifactorial disease. This evidence highlights the complex and polyfactorial nature of food addiction and prompted us to study the transcriptomic changes underlying

the resilient and vulnerable phenotype to develop food addiction-like behavior. The comparison of the transcriptomic profiles in the mPFC between addicted and non-addicted mice revealed a remarkable upregulation of the *Drd2*, *Adora2A*, *Gpr88*, and *Drd1* genes in addicted mice independently of the genotype. All these genes encode G-protein-coupled receptors and are known to be involved in the neurobiological pathways recruited in addiction[40]. $D_2R$ and $A_{2A}R$, encoded by *Drd2* and *Adora2A*, respectively, are colocalized in the mPFC glutamatergic neurons projecting from the L2/3 to L5 at presynaptic terminals[41]. Here, receptor activation synergistically inhibit synaptic glutamatergic transmission[41]. This is contrary to the well stablished postsynaptic antagonistic interaction in the GABAergic MSNs of the striatum[42]. Regarding *Gpr88*, inactivation of this gene enhanced the excitability of both D1 and D2 MSNs in the striatum[43]. Therefore, we could speculate that upregulation of Gpr88 in the mPFC of addicted mice could also contribute to the alteration of cortico-limbic system in these mice. On the other hand, $D_1R$ (encoded by *Drd1*) in the striatum conforms the direct pathway (GO), which promotes the approach to addiction-like behavior opposite to the $D_2R$ stimulating the indirect pathway (NO GO)[6]. In contrast, $D_1R$ stimulation in the mPFC decreases release of glutamate onto L5 pyramidal cells[44] similar to $D_2R$ does[41]. Thus, the increased *Drd1* mRNA levels in PFC of the addicted mice could also contribute to the decreased excitability of the glutamatergic mPFC neurons projecting to NAc. Therefore, it could be of interest to explore in future studies whether overexpression of $D_1R$ in mPFC can also modify the development of food addiction-like behavior.

Notably, the most differentially expressed gene found in our study was the *Drd2*. To our knowledge, this is the first study revealing an increased expression of the gene encoding for $D_2R$ in the mPFC in the context of addiction. In contrast, a reduced levels of $D_2R$ in the striatum have been already implicated in this disorder[45]. Thus, neuroimaging studies reported a downregulation of $D_2R$ in the striatum, which correlated with an hypofunction of the PFC in cocaine abusers[45]. In agreement, a specific *Drd2* polymorphism is associated with the "Reward Deficiency Syndrome", consisting in a hypodopaminergic state due to the altered $D_2R$ function[46].

In addition, several genes in the mPFC were downregulated in addicted mice independently of the genotype. Interestingly, genes related to neuronal plasticity (*Myh11*, *Acta2*, and *Cdh1*)[24,25], prostaglandin synthesis (*Ptgds*)[26] and gene regulation (*Fosb*)[27] were differentially expressed. The transcriptomic analysis between WT and Glu-CB1-KO mice revealed a downregulation in the *Cnr1* gene as expected in the mPFC of the mutants confirming the genetic deletion of the $CB_1R$ in this area. Additionally, *Fos* gene, encoding for c-fos protein, was less expressed in the mutants implicating an enhanced neuronal activity in mPFC of WT compared to Glu-CB1-KO mice.

In agreement to our transcriptomic data analysis, we predicted that the upregulation of *Drd2* gene expression in PL-NAc core pathway could have a critical role in promoting the vulnerability to develop food addiction. Importantly, ISH experiments demonstrated endogenous *Drd2* mRNA expression in the PL of naive WT mice, although at very low levels close to the limit of detection. Therefore, we assume that addicted mice may have a significant increase in endogenous *Drd2* mRNA levels based on gene expression data. Then, we aimed at mimicking the upregulation of *Drd2* gene expression found in the mPFC of addicted mice by overexpressing $D_2R$ selectively in the PL-NAc core projections. Our results revealed that these mice showed enhanced compulsive eating behavior despite the aversive consequences. Previous studies have demonstrated that the PL-NAc core projections are regulated by presynaptic $D_2R$ and directly innervate the $D_2R$ expressing MSNs (D2-MSNs) of the

indirect pathway[47]. Additionally, the inhibition of the D2-MSNs in the NAc core enhanced motivation for cocaine in a self-administration paradigm, but not for standard food[48]. Therefore, our results provide a new mechanism of the loss of inhibitory control for food seeking behavior involving $D_2R$ in PL cortical projections to NAc core. In particular, overexpression of $D_2R$ diminished the excitability of the pyramidal neurons in the PL projecting to NAc core. According to our model (Supplementary Fig. 11b), we predict that reduced glutamatergic transmission in the NAc core will decrease the activation of D2-MSN indirect pathway, thereby suppressing the avoidance behavior (NO GO response) and promoting the loss of control towards palatable food consumption.

In summary, we elucidated the crucial role of the glutamatergic PL-NAc core pathway modulated by $CB_1R$ and $D_2R$ as a critical mechanism for the loss of inhibitory control for palatable food seeking and consumption. An increase in the activity of this pathway plays a key role in resilience to develop food addiction. Our results provided new mechanistic underpinnings of food addiction, which might be valued for other psychiatric disorders with alterations in compulsive behavior due to the transdiagnostic nature of this concept. In addition, our findings pave the way for prevention measures, by identifying mechanisms required for strengthening the resilient phenotype.

## Methods

**Behavioral experiments**. Self-administration session: The beginning of each self-administration session was signaled by turning on a house light placed on the ceiling of the chamber during the first 3 s. Daily self-administration sessions maintained by chocolate-flavored pellets lasted 1 h in the long food addiction protocol and 2 h in the short food addiction protocol to increase the exposure of the palatable pellets on each day to ensure the development of the addiction-like phenotype. The self-administration sessions were composed by two pellet periods (25 min and 55 min) separated by a pellet-free period (10 min). During the pellet periods, pellets were delivered contingently after an active response paired with a stimulus light (cue light). A time-out period of 10 s was established after each pellet delivery, where the cue light was off, and no reinforcer was provided after responding on the active lever. Responses on the active lever and all the responses performed during the time-out period were recorded. During the pellet-free period, no pellet was delivered and this period was signaled by the illumination of the entire self-administration chamber. In the operant conditioning sessions, mice were under fixed ratio 1 (FR1) schedule of reinforcement (one lever-press resulted in one pellet delivery) followed by an increased FR to 5 (FR5) (five lever-presses resulted in one pellet delivery) for the rest of the sessions. As previously described[49], the criteria for the achievement of the operant responding were acquired when all of the following conditions were met: (1) mice maintained a stable responding with <20% deviation from the mean of the total number of reinforcers earned in three consecutive sessions (80% of stability); (2) at least 75% responding on the active lever; and (3) a minimum of 5 reinforcers per session. After each session mice were returned to their home cages.

Three addiction-like criteria: Three behavioral tests were used to evaluate the food addiction-like criteria as recently described[5] and adapted from cocaine addiction-like in rats[50]. These three criteria summarized the hallmarks of addiction based on DSM-IV[50], specified in DSM-5 and now included in the food addiction diagnosis through the YFAS 2.0[3].

Persistence to response: Non-reinforced active responses during the pellet-free period (10 min), when the box was illuminated and signaling the unavailability of pellet delivery, were measured as a persistence of food-seeking behavior. On the 3 consecutive days before the progressive ratio mice were scored.

Motivation: The progressive ratio schedule of reinforcement was used to evaluate the motivation for the chocolate-flavored pellets. The response required to earn one single pellet escalated according to the following series: 1, 5, 12, 21, 33, 51, 75, 90, 120, 155, 180, 225, 260, 300, 350, 410, 465, 540, 630, 730, 850, 1000, 1200, 1500, 1800, 2100, 2400, 2700, 3000, 3400, 3800, 4200, 4600, 5000, and 5500. The maximal number of responses that the animal performs to obtain one pellet was the last event completed, referred to as the breaking point. The maximum duration of the progressive ratio session was 5 h or until mice did not respond on any lever within 1 h.

Compulsivity: Total number of shocks in the session of shock test (50 min) performed after the PR test, when each pellet delivered was associated with a punishment, were used to evaluate compulsivity-like behavior, previously described as resistance to punishment[5,50]. Mice were placed in a self-administration chamber without the metal sheet with holes and consequently with the grid floor exposed (contextual cue). In this shock-session, mice were under a FR5 schedule of reinforcement during 50 min with two scheduled changes: at the fourth active

lever-response mice received only an electric footshock (0.18 mA, 2 s) without pellet delivery and at the fifth active lever-response, mice received another electric footshock with a chocolate-flavored pellet paired with the cue light. The schedule was reinitiated after 10 s pellet delivery (time-out period) and after the fourth response if mice did not perform the fifth response within a min.

Attribution of the three addiction-like criteria: After performing the three behavioral tests to measure the food addiction-like behavior, mice were categorized in addicted or non-addicted animals depending on the number of positive criteria that they had achieved. An animal was considered positive for an addiction-like criterion when the score of the specific behavioral test was above the 75th percentile of the normal distribution of the chocolate control group. Mice that achieved two or three addiction-like criteria were considered addicted animals and mice that achieved 0 or 1 addiction-like criteria were considered non-addicted animals.

**Surgery and virus vector microinjection**. General surgical procedures: Mice were anesthetized as reported in the drugs section (Supplementary Information) and placed into a stereotaxic apparatus for receiving the adeno-associated virus (AAV) intracranial injections. All the injections were made through a bilateral injection cannula (33-gauge internal cannula, Plastics One, UK) connected to a polyethylene tubing (PE-20, Plastics One, UK) attached to a 10 μl microsyringe (Model 1701 N SYR, Cemented NDL, 26 ga, 2 in, point style 3, Hamilton company, NV). The displacement of an air bubble inside the length of the polyethylene tubing that connected the syringe to the injection needle was used to monitor the micro-injections. The volume [0.2 μl per site in prelimbic (PL), 0.4 μl per site in nucleus accumbens core (NAc core)] was injected at a constant rate of 0.05 μl/min (PL) or 0.1 μl/min (NAc core) by using a microinfusion pump (Harvard Apparatus, Holliston, MA) for 4 min. After infusion, the injection cannula was left in place for an additional period of 10 min to allow the fluid to diffuse and to prevent reflux, and then it was slowly withdrawn during 10 additional min. We used the following coordinates to target our injections according to Paxinos and Franklin:[51] (PL) AP + 1.98 mm, $L \pm 0.3$ mm, DV − 2.3 mm; (NAc core) AP + 1.34 mm, $L \pm 1$ mm, DV − 4.6 mm.

Viral vectors: We used the following vectors: AAV-hM4Di-DREADD (AAV8-hSyn-DIO-hM4D(Gi)-mCherry, 1.21E + 13 gc/ml), AAV-control-DREADD (AAV8-hSyn-DIO-mCherry, 1.19E + 13 gc/ml) and AAV-retrograde-Cre-GFP (AAVrg-Syn1-GFP-Cre; 8.2E + 12 gc/ml) from Viral Vector Production Unit of Universitat Autònoma de Barcelona, AAV-retrograde-Cre-BFP (AAVrg pmSyn1-EBFP-Cre; $6 \times 10^{12}$ vg/ml) from Addgene (viral prep # 51507-AAVrg), AAV-$D_2$R (AAV2/1-hSyn-DIO-SF-$D_2$R(L)-IRES-mVenus,1,23E + 13 gc/ml), the plasmid was a gift from Christoph Kellendonk and Jonathan Javitch's lab[52], and the corresponding AAV-control (AAV1/2-hSyn-floxstop-hrGFP, 7,69E + 11 gc/ml) was from Beat Lutz's lab. For the inhibition of glutamatergic neurons in the PL subregion, a bilateral injection targeting the PL was performed in Nex-Cre mice. Mice received an injection of 0.2 μl per site of the AAV-hM4Di-DREADD or 0.2 μl per site of the AAV-control-DREADD. For the specific inhibition of the projecting neurons from PL to NAc core, two bilateral injections were performed in WT C57BL/6J mice, one targeting the PL and the other the NAc core. Mice received an injection of 0.2 μl per site of the AAV-hM4Di-DREADD into PL and an injection of 0.4 μl per site of the AAV-retrograde-Cre-EBFP into the NAc core. A subset of mice used for electrophysiological recordings in the NAc received the AAV-retrograde-Cre-GFP. For the overexpression of $D_2$R in the PL-NAc core projecting neurons, two bilateral injections were performed in WT C57BL/6J mice, one targeting the PL and the other the NAc core. Mice received bilateral injections of 0.2 μl of the AAV-$D_2$R or 0.2 μl per site of the AAV-control into PL and bilateral injections of 0.4 μl of the AAV-retrograde-Cre-EBFP into the NAc core.

To detect the viral expression in all the experiments we visualized each reporter as it is listed below:

For the detection of AAV8-hSyn-DIO-hM4D(Gi)-mCherry and AAV8-hSyn-DIO-mCherry in Nex-cre mice, we directly visualized mCherry in the confocal microscope. mCherry is a bright red monomeric fluorescent protein that was clearly visible in our experimental conditions without performing an immunofluorescence.

For the detection of AAV8-hSyn-DIO-hM4D(Gi)-mCherry and AAV8-hSyn-DIO-mCherry in PL, and AAVrg-pmSyn1-EBFP-Cre targeting the NAc, we performed an immunofluorescence immunocytochemistry against Cre to detect the correct viral infection in the NAc and visualized the labeled Cre and the mCherry in the confocal microscope.

For the detection of AAV2/1-hSyn-DIO-SF-D2R(L)-IRES-mVenus and AAV1/2-hSyn-floxstop-hrGFP in the PL and AAVrg-pmSyn1-EBFP-Cre targeting the NAc, we performed an immunofluorescence immunocytochemistry against mVenus/GFP and against Cre to detect the correct viral infection in both brain areas. We visualized the labeled mVenus or GFP, and the labeled Cre in the confocal microscope. We performed a double immunohistochemistry analysis using anti-$D_2$R and anti-mVenus/GFP antibodies to visualize the expression of $D_2$R at post- and presynaptic terminals.

**Experimental design**. In the first experiment (Fig. 1a), Glu-CB1-KO mice ($n = 58$) and WT mice ($n = 56$) were trained under FR1 schedule of reinforcement during six sessions, followed by 112 sessions of FR5 to self-administer chocolate-flavored

pellets. The three addiction-like criteria (1) persistence to response (2), motivation (3), and compulsivity were evaluated at two different time points in each mouse. The first time point was the early period (sessions 1–18 of FR5) and the second time point was the late period (sessions 95–112 of FR5). For the in vitro electrophysiological recordings, we used naive Glu-CB1-KO mice ($n = 6$) and their WT littermates ($n = 6$) (see section In vitro electrophysiology in brain slices).

For the inhibition of the glutamatergic transmission in PL subregion (Supplementary Fig. 4e), mice followed the same behavioral procedure described for the early period in the first experiment with some variations due to the surgical AAV injection. In particular, Nex-Cre mice were trained to acquire the operant conditioning maintained by chocolate-flavored pellets under FR1 (two sessions) and FR5 (two sessions) schedule of reinforcement followed by the surgery for injecting Cre-dependent AAVs carrying the DREADD (DREADD approach). After bilateral intracranial injection of the AAV-hM4Di in the PL, the expression of the AAV was allowed during the period of four weeks. At the beginning of this period, mice were under FR5 (four sessions) to recover the basal levels of operant responding. At the end of these four weeks, an osmotic minipump filled with CNO or saline was subcutaneously implanted in the back of each mouse. Subsequently, during the chronically CNO-induced activation of the expressed hM4Di receptors, mice were under FR5 scheduled sessions followed by the measurement of the three addiction-like criteria. To verify the viral expression, mice were perfused at the end of the experiment, and the fluorescent reporter mCherry was visualized in brain slices using a Leica DMR microscope equipped with a digital camera Leica DFC 300FX (10x objectives). In a subset of injected mice, in vitro electrophysiology recordings were used to verify CNO-induced suppression of neuronal activity (see section In vitro electrophysiology in brain slices).

For the specific inhibition of the projecting neurons from PL to NAc core (Fig. 3g), mice followed a similar experimental design as described above with a modification in the surgical intervention. In this experiment, a dual viral approach was performed to selectively silence the PL neurons that project to NAc core (retro-DREADD approach): bilateral intracranial injection of AAV-hM4Di targeting the PL and of AAV-retrograde-Cre targeting the NAc core. To verify viral expression, mice were perfused at the end of the experiment and the fluorescent reporter mCherry was visualized in brain slices, as previously described. Cre-recombinase expression was detected by immunofluorescence using an anti-Cre recombinase antibody (see section Immunofluorescence studies in Supplementary Information). Same as in the previous experiment, in vitro electrophysiology recordings were used in a subset of injected mice to verify CNO-induced suppression of neuronal activity (see section In vitro electrophysiology in brain slices).

For the transcriptomic analysis, Glu-CB1-KO and WT mice were sacrificed immediately after the last FR5 session of the food addiction procedure and mPFC was extracted by macrodissection to perform RNA sequencing (see section RNA sequencing).

For the overexpression of $D_2$R in PL-NAc core pathway (Fig. 5g), mice followed the same behavioral and surgical procedure with a dual vector approach similar to that described in the previous experiment with slight modifications: (1) AAV-$D_2$R or AAV-control was injected in the PL, (2) the surgical intervention for the osmotic minipump filled with CNO was not required, and (3) an immunofluorescence assay was performed after the perfusion of the mice using an anti-$D_2$R and anti-GFP antibody that visualize the overexpressed $D_2$Rs and mVenus reporter of the AAV injected in PL and against Cre recombinase to visualized the injection site of the retrograde AAV in the NAc core and the retrograde transport to the PL (see section Immunofluorescence studies in Supplementary Information). Equally to previous experiments, in vitro electrophysiology recordings were used in a subset of injected mice to verify that overexpression of $D_2$R induced the suppression of neuronal activity by using $D_2$R agonists, quinpirole and dopamine (see section In vitro electrophysiology in brain slices).

**In vitro electrophysiology in brain slices**. Animals were sacrificed and brains were quickly removed obtaining coronal slices (300 μm) with vibratome (Leica VT1200S) upon the presence of low-$Na^+$ cutting solution (composition in mM: Sucrose 212, NaHCO3 27, Dextrose 10, CaCl2 2.2, MgSO4 2.2, KCl 2, NaH2PO4 1.5; pH 7.4 when saturated with 95% $O_2$ + 5% $CO_2$). Afterwards, slices were incubated (40 min/34 °C) in artificial cerebrospinal fluid (ACSF; composition in mM: NaCl 124, KCl 2.5, NaHCO3 26, CaCl2 2, MgCl2 1, NaH2PO4 1.25, glucose 10; pH 7.4 when saturated with 95% $O_2$ + 5% $CO_2$). Visualization of brain slices were performed with an upright microscope (BX51WI, Olympus), outfitted with x4 lens, x40 water immersion lens, Nomarsky optics and mercury lamp with adequate filters for blue (470–490 nm) and green (533–580 nm) light stimulation. L5 pyramidal neurons were recognized by their position along the cortical column, soma shape and presence of apical dendrite and electrophysiological properties.

Voltage-clamp and current-clamp electrophysiological recordings were performed by using a Multiclamp 700B amplifier (Axon Instruments), filtered at 1–2 kHz and digitized at 20 kHz with a 16 bits Axon Digidata 1550B (Axon Instruments). Protocols design and data acquisition were performed with pClamp9.2 software (Axon Instruments). Borosilicate patch pipettes (1.5 mm o.d., 0.86 mm i.d., with inner filament; Harvard Apparatus) were used after pulled (P-97, Sutter Instrument). Pipette resistance was calculated with pClamp software and was estimated among 8–10 MΩ. Electrical DC pulses were applied with a DS3

Isolated Stimulator (Digitimer) using a theta-glass pipette filled with ACSF solution. All recordings were performed at RT (21–23 °C).

Paired-Pulse facilitation recordings in layer 5 of Glu-CB1-KO mice: Synaptic facilitation was achieved by applying consecutive electrical stimulus with a 50 ms interpulse interval in L2/3 of mPFC. The evoked field postsynaptic potentials (fPSP) were recorded in the L5 with ACSF filled patch pipette. Once the evoked fPSP were stabilized, at least 50 consecutive responses were recorded. Synaptic facilitation was estimated as the ratio among second response (P2) respect the first response (P1)[41]. Electrical stimulus (0.06 Hz) was the 50% of the intensity needed to evoke the maximal fPSP.

Layer 5 evoked fPSP pharmacological modulation: A single fPSP was evoked on L5 by stimulating on L2/3 as previously described. Once fPSP was stable, 50 consecutive responses were recorded to stablish the baseline fPSP amplitude. Each pharmacological application was perfused on the recording chamber for 20 min while recording evoked fPSP. The effect of WIN55,212-2 5 μM (Sigma-Aldrich, Spain) and rimonabant 4 μM (Sanofi-Aventis, Spain) in the fPSP amplitude was calculated by averaging the last 10 evoked fPSPs.

Pharmacological modulation of L5 pyramidal neurons properties of mPFC: Somatic current-clamp whole-cell recordings were obtained in the presence of ACSF and pharmacological treatment. Intracellular solution composition (in mM) was: KMeSO4 135, KCl 10, HEPES 10, NaCl 5, ATP-Mg 2.5, GTP-Na 0.3; pH adjusted to 7.3 by adding KOH. Hyperpolarizing and depolarizing square current pulses were applied (from −200 to 300 pA; Δ25 pA; 1 s duration). Resistance was obtained from the first depolarizing pulse. Rheobase was calculated by applying a ramp hyperpolarizing current-pulses (from 150 to 300 pA; 1.5 s duration). hM4Di-mCherry-positive neurons modulation was estimated after 10 min of perfusing recording chamber with CNO 10 μM (Enzo Life Sciences, NY) diluted in ACSF. Dopamine hydrochloride 10 μM (Sigma-Aldrich, Spain) and quinpirole hydrochloride 2 μM (Sigma-Aldrich, Spain) modulation of $D_2$R-mVenus and GFP-positive neurons was performed after 10 min perfusing recording chamber.

mEPSC and mIPSC detection in PL mPFC and NAc of Glu-CB1-KO mice: Patch recording pipettes (4–9 MΩ) were filled with 115 mM cesium methanesulfonate, 20 mM CsCl, 10 mM Hepes, 2.5 mM MgCl2, 4 mM Na2ATP, 0.4 mM Na3GTP, 10 mM sodium phosphocreatine, and 0.6 mM EGTA, pH 7.25. For mIPSCs recordings, cesium methanesulfonate was replaced with the same concentration of CsCl.

Recordings of miniature responses (mEPSC and mIPSC) were performed using the whole-cell patch clamp technique in neurons at −70 mV in PL and at −75 mV in NAc in the presence of 1 μM tetrodotoxin, 50 μM picrotoxin, and 100 μM saclofen (for mEPSCs) or 50 μM AP5 and 20 μM CNQX (for mIPSCs). Evoked synaptic responses were evoked with bipolar electrodes using single-voltage pulses (200 μs, up to 20 V). Stimulating electrodes were theta capillaries (4–9 μm tip) filled with ACSF. The stimulating electrodes were placed over NAc core 100–150 μm from the recorded cells. Electrophysiological recordings and data acquisition were performed with PC-ONE amplifiers and pClamp software (Molecular Devices). Analysis was performed with pClamp software (miniature currents) or custom-made Excel (Microsoft) macros.

Pharmacological modulation of MSNs properties of NAc: The effects of pharmacological agents were tested at 10–15 min after bath application of WIN55,212-2, 5 μM, CNO 10 μm, or quinpirole hydrochloride 2 μM, where appropriate, using whole-cell mode and were quantified using 5 min bins.

**RNA sequencing**. RNA extraction: Tissue collection was performed immediately after the last FR5 session. After decapitation, the brains were removed from the skulls and processed rapidly on ice. The mPFC was isolated by macrodissection according to the following coordinate from Paxinos and Franklin 2001[51] (AP + 1.98 mm) and the samples were placed in individual tubes, frozen on dry ice, and stored at −80 °C until RNA isolation for the RNA sequencing. The remaining brain parts from the same animals were frozen on dry ice and stored at −80 °C.

Total RNA from mPFC was extracted using a miRNeasy Mini kit for subsequent RNA-seq analysis and RT-PCR validation. Briefly, tissues were homogenized in QIAzol Lysis Reagent and, after adding chloroform, the aqueous phase was collected and microRNA and total RNA were extracted by using miRNeasy Mini kit (Qiagen, 217004).

Library preparation and total RNA sequencing: Further evaluation of the RNA, RNA library generation, and sequencing were carried out by StarSEQ GmbH (Mainz, Germany). Sequencing was performed on an Illumina NextSeq 500 sequencer with High Output chemistry and minimum output of 40 million single end reads (75 bp) per sample.

Total RNA (750–1000 ng/sample) was rRNA depleted by Ribo-Zero rRNA Removal Kit (Human/Mouse/Rat) from Illumina. For sequencing library generation, the NEBNext Ultra II Directional RNA Library Kit was used. Resulting RNA libraries were size selected to a median insert size of 300 bp.

RNAseq data analysis: RNA sequencing for WT and Glu-CB1-KO mice were performed under addicted and non-addicted conditions. RNA-seq output was received as raw files in fastq format. To check the quality of individual sequenced sample, we used FASTQC version v0.10.5. Quality check followed by alignment using TopHat version v2.1[53]. to the mouse genome (mm9) with default parameters. Further, mapped reads were considered for read count per gene using HTSeq version 0.9[54]. HTSeq output (read counts per gene) were normalized and

differential gene expression analysis was performed using R package DESeq[55] with FDR rate of 0.1. Variability between the addicted and non-addicted mice was determined using PCA analysis from "plotPCA" function of DEseq package. For PCA analysis, top varying 500 genes were selected. P-value were calculated using nbinomTest function from DEseq package.

Differential expression analysis was performed between addicted and non-addicted mice. Similarly, we performed differential expression analysis between WT and Glu-CB1-KO. All genes with a fold change higher than >1.5-fold, P-value < 0.01 and average read counts > 40 in either condition comparison (i.e., addicted vs. non-addicted and WT vs. KO mice) were selected as differentially expressed genes. Further, it was visualized with volcano plot using R. The RNA sequencing data used in this study is available at GEO under accession number GSE139482.

**Statistical analysis**. All statistical comparisons were performed with SPSS (IBM, version 25). Comparisons between two groups were analyzed by Student t-test or U Mann–Whitney and within groups by paired t-test or Wilcoxon test depending on the distribution defined by the Kolmogorov-Smirnov normality test. ANOVA with repeated measures was used when required to test the evolution over the time. Two-way ANOVA by subsequent post hoc analysis (Fisher PLSD) was used for multiple group comparison. The Pearson correlation coefficient was used to analyze the relationship between values in each addiction-like criteria and the final criteria achieved. The chi-square analyses were performed to compare the percentage of addicted mice with the non-addicted ones, we compared the observed frequencies with the frequencies obtained in the control group. Results were expressed as individual values with the median and the interquartile range or with the mean ± S.E.M specified in the figure legend. A P-value < 0.05 was used to determine statistical significance. The sample size was calculated based on the power analysis. The criterion for significance (alpha) was set at 0.050 and the statistical test used was two-sample t-test. With the sample size of 12–22 mice per group, our studies achieved a power between 73 and 90%. Supplementary tables (Supplementary Tables 1–6) provided a complete report of the statistical results for the data described in the figures.

See Supplementary Information for extended details for animals, operant behavior apparatus, food pellets, drugs, RT-PCR validation, immunofluorescence studies and fluorescence in situ hybridization.

**Reporting summary**. Further information on research design is available in the Nature Research Reporting Summary linked to this article.

## Data availability

Individual data points are graphed in main and Supplementary Figures. All relevant data that support this study are available from the corresponding author to any interested researcher upon reasonable request and at https://doi.org/10.6084/m9.figshare.11366015. The RNA sequencing data used in this study is available at GEO under accession number GSE139482.

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

## Acknowledgements

We thank A. Domi, E. James, E. Cuboni, R. Martín, M. Linares, F. Remmers, C. Maul, A. Conrad, and D. Real for their help during the experiments. The support by the IMB Microscope Core Facility (Mainz, Germany) is gratefully acknowledged. C. Kellendonk and J. Javitch (Columbia University, New York, USA) provided the AAV-hSyn-DIO-D2L-mVenus construct. This work was supported by the Spanish Ministerio de Economía y Competitividad-MINECO (#SAF2017-84060-R-AEI/FEDER-UE), the Spanish Instituto de Salud Carlos III, RETICS-RTA (#RD12/0028/0023), the Generalitat de Catalunya, AGAUR (#2017 SGR-669), ICREA-Acadèmia (#2015) and the Spanish Ministerio de Sanidad, Servicios Sociales e Igualdad, Plan Nacional Sobre Drogas (#PNSD-2017I068) to R.M., Fundació La Marató-TV3 (#2016/20-30) to E.M.-G., the German Research Foundation (#CRC1193 "Neurobiology of Resilience", TP A05 and B04) to B.L. and S.G., and the Boehringer Ingelheim Foundation to B.L., S.G. and I.R.A. The work of M.N.A was supported by the Emergent AI Center funded by the Carl-Zeiss-Stiftung. NARSAD Young Investigator Award (#22434), MINECO Ramón y Cajal (#RYC- 2014-15784) and (#SAF2016-76565-R) and Fondo Europeo de Desarrollo Regional (FEDER) to R.A. DIUE Generalitat de Catalunya (#2017 SGR 595), MINECO (#SAF2016-79956-R), EU (#Era Net Neuron PCIN-2013-060), Fundació La Marató-TV3 (#2016/20-31), CRG acknowledges support of the Spanish Ministry of Economy, Industry and Competitiveness (MEIC) to the EMBL partnership, the Center of Excellence Severo Ochoa (#SEV-2012-0208), the CERCA Programme/Generalitat de Catalunya, the CIBER of Rare Diseases of the ISCIII to M.D. MINECO Ramón y Cajal (#RYC-2016-20414), AGAUR (#2017 SGR 926), (#RTI2018-094887-B-I00) and Fondo Europeo de Desarrollo Regional (FEDER) to M.N.

## Author contributions

E.M.-G., L.D.-R. and R.M. conceived and designed the behavioral studies with input from B.L. and I.R.A.; B.L. generated the conditional transgenic mice (Glu-CB1-KO); L.D.-R. performed the behavioral experiments and the statistical analyses and graphs with the supervision of E.M.-G. and R.M.; S.K. collaborated with the extraction of the samples; R.A., R.M. and E.M.-G. designed the chemogenetic experiment; E.M.-G. and R.M. set up the DREADD and retro-DREADD approach; L.D.-R., E.S. and I.R.A. performed chemogenetic surgeries; L.D.-R., E.S. and I.R.A. performed immunofluorescence studies and the confocal images supervised by E.M.-G., R.M. and B.L.; E.D., I.S. and M.N. performed and designed electrophysiology experiments supervised by M.D, M.N., R.M. and E.M.-G.; I.R.A. and M.N.A. performed the RNA sequencing and the qRT-PCR experiments and the bioinformatic analysis supervised by B.L. and G.S.; C.H., S.B. and I.R.A. performed FISH assay; E.M.-G., L.D.-R., B.L. and R.M. wrote the manuscript and I.R.A provided critical review of the manuscript with inputs from all the other authors.

## Competing interests

The authors declare no competing interests.
