## [Peer Review File · Nature Communications]

Reviewers' comments:

Reviewer #1 (Remarks to the Author):

This study explores the pathways and mechanisms that drive addictive/compulsive behaviors towards highly palatable food. The study identifies the prelimbic (PL) cortical area of mice and their projections to the nucleus accumbens as important in exerting inhibitory control over food intake in regards to the development of perseverative responding, motivation and punishment resistant taking. Using cell specific knockout mice for CB1R and manipulations of neuronal excitability using chemogenetic and *Drd2* overexpression in the PL, the study shows that strengthening of the PL-NAC pathway promotes resilience while decreasing the excitability of the PL-NAC neurons generates vulnerability for compulsive taking of highly palatable food.

This is a rational, thoughtful and well-done study. It is clearly written and the logic and rationale of experiments is easy to follow. Most important methodological information is included and the conclusions reached are logical and not overstated for the most part (but see below). Appropriate controls are performed and shown (but see one more suggested).

I have a few comments and a suggestion for a control that I think that can improve the clarity and extend the relevance by highlighting the selectivity of the approach.

Main comments:

1. There is an important control for the loss of CB1R modulation of glutamatergic transmission in the cortex of the Glu-CB1 KO mice, which targets telencephalic glutamatergic neurons. This is shown in Figure 2 and suppl. Fig too. Now, inclusion of a control for the selectivity of the manipulation would be also very important. For example, is there still modulation of GABAergic transmission in the cortex of these KO mice. Knowing this would allow the authors to further pin point the behavioral effects as being caused by that loss of glutamatergic transmission modulation, which to my view, it is implied there but not really shown (see other two points below as well).
2. The text seems to put too much emphasis on the glutamatergic transmission in the cortex. It is true that this is one of the factors altered in the CB1R KO mice but in general the study seems to show that it is actually the overall activity and the engagement of the L5 pyramidal neurons in the PL cortex and their projections to the accumbens which are important on controlling addictive behaviors towards palatable food. So the study will benefit from revising the text and the discussion to represent this.
3. There are several occasions in which the cells losing CB1R and the cells recorded and showing the effect are being confused or presented in a very confusing matter. This is especially true in the discussion section (page 19), where the pyramidal neurons lacking CB1R are recorded but the changes measured are unlikely due to the loss of the CB1R on these cells but rather on the presynaptic glutamatergic neurons that form synapses onto them and where CB1R presynaptically inhibits glutamate release.
4. What is the body weight of Glu-CB1 KO mice and how it compares to WT? how could the body weight information affect the interpretation of the findings with regards to lower motivation and compulsivity?
5. It would help to sharpen the terminology of the ephys mini data. The results section refers to

"spontaneous" release but these are miniature responses obtained in TTX, then the authors could skip the spontaneous term and also maybe make inference on release probability.

Minor comments:

6. Page 17, last sentence of the result section is a bit confusing or inaccurate. In the sense that the D2R do not decrease the excitability but rather dopamine via D2rs. It could be revised.

7. Figure 6, panel A. Wrong labels. They are the same as figure 5 and should be fixed to reflect the Drd2 overexpression manipulation.

8. DREADD figures. Very nice that study includes and shows the controls showing the manipulations work as expected. Could the authors move to main figure the control of CNO application in the absence of the hM4Di expressing to the main figure. At least for the ephys parameter shown in the figure. (others could stay in the suppl. Figure).

9. Page 18- "synaptic transmission of pyramidal glutamatergic neurons" might be more clear if it reads "on" or "onto"

10. Page 19. "decrease selectively the glutamatergic activity of the PL region" is confusing.

11. Personally, statements such as "for the first time..." do not add much.

Veronica A Alvarez

Reviewer #2 (Remarks to the Author):

Domingo-Rodriguez and collaborators report an impressive set of results indicating that CB1 cannabinoid receptors expressed by glutamatergic neurons in the median prefrontal cortex regulate the development of addictive-like behaviors in a mouse model of 'food addiction'. They further show that this response is at least partly mediated by mPFC projections to the nucleus accumbent, and that the development of 'food addiction' is accompanied by profound changes in the expression of dopamine D2 receptors in the mPFC. The experiments are conducted with the expertise and quality that is the norm for these excellent investigators. The manuscript is clear and well written. The interpretation of the data is sound, though at time hyped and incomplete (as explained below).

1. A first conceptual issue is the apparently acritical acceptance of food addiction as a construct. I am agnostic on this point, but given the present controversy i recommend that the authors do a better job at positioning their study. The statement that 25% of the world population is affected by food addiction, taken from a review, is implausible.

2. Another conceptual issue pertains to the mouse model used in the study. Previous work has shown that removal of CB1R from telencephalic glutamatergic neurons has a marked impact on food intake. What is the possible contribution of this trait to the changes observed? A deeper discussion is needed about this point.

3. I would recommend increasing the number of samples in the physiological experiment of Figure 2b.

As it stands, the claimed difference can be ascribed to only two samples, and is not credible.

4. The transcriptome analyses indicate a large elevation in *Drdr2* in 'addicted' versus 'non-addicted' mice. This is really not a great surprise, given the role of this receptor in motivated behavior. But the analyses also show an elevation in *Gpr88*, which is a lot more intriguing and potentially novel. Incidentally, the names of genes whose expression is reduced should also be listed, and discussed.

5. I would be careful in the use of the word 'addicted' etc. A key component of addiction is the persistence in a behavior that is recognized as detrimental. There is no such recognition in the mouse. Phrases like 'addiction-like' etc appear to be more appropriate in this context.

Reviewer #3 (Remarks to the Author):

In this manuscript, authors elucidate some of the mechanisms underlying food addiction “vulnerability” by using KO animals, chemogenetics, gene overexpression experiments and performing behavioural, electrophysiological and RNA seq analysis.

Authors show that CB1 deletion in dorsal telencephalic glutamatergic neurons changes food addiction profile. They then show that mPFC-NAC core projections silencing by DREADDs increases compulsive food seeking. RNAseq revealed that D2R was upregulated in the mPFC of addicted animals, so authors overexpressed D2R in this brain region, which promoted food addiction phenotype.

Some of the results of the manuscript are very interesting and novel, and experimental design was performed adequately and with state of the art tools. However, I think that there some limitations that require attention. Please see my comments below, I hope it helps authors to improve the manuscript and strengthen their conclusions.

As a general remark, I missed coherence and integration of the different parts of the manuscript. In the beginning that is not completely clear why authors include the CB1-KO data – I think this data is really nice, but in my opinion, it does not add much to the other part of the story regarding D2R (and vice versa) (and similar CB1-KO data was previously published by the team – NPP 2015). The two parts lack integration (authors also appear to partially agree with this, as they state in page 15 “we hypothesized that this (D2R) upregulation could play a key role in the development of food addiction behavior, irrespective of the presence or absence of CB1R.”

Yes, both endocannabinoid and dopaminergic control of PL neurons modulate food addiction, but how do they interact (if they do)? If it is just a matter of modulation of PL-NAC glutamatergic signalling (as I think authors postulate), then, for example, you could register the activity of striatal D2-MSNs (and D1-MSNs) in glu-CB1KO to show that glutamatergic signalling is indeed increased in the NAc NO GO pathway, as authors hypothesize. Then block it to see if the resilient phenotype is abolished?

One other caveat of the study is that ephys recordings were always made in the L5 and never in the NAc, so it is risky to assume that the resilient and vulnerable phenotype arises from this particular connection, as authors assume. Both CB1-KO and D2R overexpression experiments of this study do not exclude the contribution of other brain regions other than NAc for the phenotype. For example, glu-CB1KO can have effects both at L5 level or presynaptically at PL terminals in the NAc core; the same is true for D2R overexpression, which can occur both at PL/L5 levels or terminals in the NAc. Some degree of PFC-NAc collateralization exists (Pinto, Sesack 2000), so this should be taken into consideration in this study and future experimental designs.

Using the same strategy as in this manuscript, If you overexpress D2R in PL-Amyg neurons for example (or other PL-connection), would you see the same effect in food addiction or not? (this could

help control if this effect is exclusive to PL-NAc core terminal overexpression or is more related to overexpression of D2RD in PL (from example). Similarly, If you knockdown D2R in PL-NAc neurons, what is the effect in the development of food addiction? The same control experiment could be performed for DREADD experiment in order to show that it is PL-NAc projections that are crucial for food addiction (and not other PL- projections). I believe these experiments would be important to disentangle the "specificity of projections" question.

Also, transcriptomic analysis is very incomplete; full data is not provided. Authors also do not show evidence of D2R overexpression.

Major

1. Statistics: Initial glu-CB1KO data is outstanding, with a large number of animals, but for some other experiments, authors did not provide G power analysis (or similar) to have a sufficient/adequate number of animals. Though one can justify that we can use "usual" number of animals as previous studies, in this work, authors are dealing with a small percentage of animals that will develop food addiction, so these calculations should be taken into consideration (see below some of my comments).
2. Fig 1b – shows infection of ACC also – this should be discussed.
3. Fig. 1b vs 3e- number of reinforcers in 1h vs 2h, respectively- why do authors do not use the same period of time?
4. Fig. 3: this figure presents some errors, and needs some clarifications.
 - a. In which layer of PL did you perform the recordings of Fig. 3c? (I assume it is L5 but this should be referred in results section).
 - b. Authors should also explain the heterogeneous firing rates of PL neurons – this is not usual. Any explanation?
 - c. There is represented only one addicted mice in Fig3f in saline group, then no addicted mice appear in 3g-h in the same saline group? Explain.
 - d. Also regarding 3i: if there is only 1/7 addicted animal in saline group, how is the percentage 12.0%? How was this calculated?
 - e. Small typo: In the table, appears 12.5%, graph appears 12%.
 - f. Authors refer that in a large population (Fig. 1), 25% of wt animals will develop food addiction; so using 8-10 animals is clearly insufficient. Was this experiment replicated in another set of animals?
 - g. Why do authors observe a decrease in persistence in CNO animals? I assume that if you would take the saline addicted animal, the two groups would be identical, which again emphasizes that the number of animals used for this experiment is insufficient.
 - h. This figure does not add much to the main conclusions of the manuscript - especially because you have Fig. 4 which is a finer approach, so I would place Fig. 3 in Sup. Material. In addition, you also have substantial transfection of the ACC brain region, which can partially contribute for the observed phenotype.
5. Fig. 4:
 - a. Page 12: "The lack of significant differences in the persistence to response and motivation highlights that the PL-NAc core projection was selectively involved in the loss of control leading to compulsive food seeking" – overstatement, please remove.
 - b. Sup. Fig 4 i-k, l-n : different number of animals in the panels and also does not match main fig 4 – why? Were animals excluded? Based on what criteria?
6. Fig 5
 - a. This data is very incomplete; authors do not provide complete list of DE genes; raw data and final analysed data should be provided. This data should also be more integrated and discussed in the paper.
 - b. what was the criteria to show those particular DE genes? Why did authors only selected upregulated

genes to be replicated by qPCR in 5e, whereas in 5h were the down-regulated which were selected?
c. fig. 5f: use two colours for the dots to distinguish between wt and CB1 KO in the distribution; 5h – color code also between addicted and non-addicted.
d. Methods regarding this experiment: depth of RNA seq? please provide more details on the sequencing procedure.
e. DRD1 is also upregulated and is known to affect PL glutamatergic activity – authors could overexpress this gene and observe if it leads to a similar/distinctive phenotype - this would be a nice control experiment to validate DRD2 overexp data

7. Fig. 6:

a. a. is wrong, please check
b. Provide evidence of D2R overexpression – qPCR or other technique – this is a major issue, as authors do not present any evidence of DR2 overex. Also, what is the correlation between D2R levels and phenotype (if any)?
c. Overexpression of D2R: are the effects mediated by overexpression at dendrites? Or pre-synaptic expression of D2R? at terminals? IHC experiments would show where is the receptor overexpressed (at both sites probably)
d. DRD2 overexpression was found in the PFC of addicted animals, a very interesting finding. Then authors overexpress D2R in PL-NAc neurons. Are the behavioural effects due to overexpression of D2R locally in the PL or overexpression in the terminals in the NAc? Having said this, it is very difficult to speak in a “selective top-down cortical pathway”. (please see my comments above)

8. Chemogenetics: Authors always register in L5 (fig. 4, 6), but then assume that the reduced PL activity is observed in less excitation of NAc core. Why did authors did not perform recordings in the NAc core to show exactly this? If one thinks this is the mechanism, then this is crucial.

9. Again, in the same line of the previous comment, The hypothesis of the vulnerable phenotype is that the overexpression of D2R in PL terminals is supposed to decrease glutamatergic signalling in NAc core – authors register PL activity, but they do not present evidence of NAc activity in response to PL stimulation in overex animals. This is crucial to show that indeed this is the “vulnerability” pathway

10. Discuss if CB1-KO effects occur mostly at PFC layers or at terminals in the NAc

11. Authors should refer if they performed IHC analysis to detect correct viral expression for all animals used in the experiments. (I assume they did, so this should be referred)

12. Why do authors opt not to include correlation data in the main figures?

13. I suggest that the Title should be changed to match the data

14. One thing that was not clear for me, but maybe I missed it, was the fact that in experiment of fig. 4 and 6, animals, test groups only present significant changes in the compulsivity score. I would expect changes also in persistence to response. What is the explanation?

Minor

1. Legend in sup fig2 is wrong. Depicted are WT animals and the “increased transmission” is not observed

2. Page 6 “Highly suppressed response” please rephrase this sentence, the difference in compulsivity is subtle. In fact, there are just some WT animals that make it go up, otherwise it would be very similar.

3. Page 6 last sentence is very confusing

4. Consider using other colour scheme throughout the manuscript– for example in sup fig 1f-h it is impossible to distinguish between black and dark blue. Red dots in red bars are also very difficult to see – e.g. Fig 4g

5. Page 9 first sentence confusing, too many claims

6. Page 9 last sentence is confusing
7. Page 11 – refer that recordings are made in L5
8. Legend fig 4. Remove “Loss of inhibitory control”, as you did not directly evaluate this parameter
9. Page 13: “Finally, a positive correlation between the number of criteria reached and the values of each addiction criterion was found in both groups (Supplementary Fig. 4i-n).” – rephrase, as in saline group the correlations are not significant for two of the tests. (the same comment for page 17)
10. Fig5f a should be *
11. page 14 -“In fact, Fos mRNA levels correlate with the increased number of reinforcers in WT as compared with Glu-CB1-KO immediately prior to sample collection (Fig. 5d).”- this correlation is not shown anywhere, remove sentence
12. page 15: “in mPFC related to addiction” – food addiction
13. Fig 6b: there is substantial background in the slice; authors should also provide better quality and magnified figures (in sup.)
14. page 15: “we confirmed low endogenous Drd2 mRNA expression in PL by in situ hybridization (ISH) (Supplementary Fig. 6a).” – compared to Cpu and NAc, I assume? No quantification is provided. A is out of place in the figure a.
15. page 15: “Cre recombinase mRNA is present in PL, and that importantly this mRNA is coexpressed with the endogenous Drd2 mRNA” -. Provide quantification (in sup is ok)
16. no effect of quinpirole in activity of L5 neurons of control animals – this is surprising, discuss (sup fig 7)
17. page 16: remove “presumably by endogenous dopamine.”
18. What is the expression of D2R in CB1KO animals?

Methods

19. name ethical committee authorization number
20. Stats: Did authors check for outliers? How did authors perform chi-square analyses – refer in methods
21. Page 14: how was the mPFC removed? Macrodissection? please refer
22. Provide more info about PCA analysis
23. DELTA DELTA CT method was used for qPCR but is unclear what was the control group? Which housekeeping genes were used for qPCR? Actin? Usp? More than one? How was the normalization done? for one of the genes or both? Please provide more information.
24. Provide serotypes for all virus used

Sup material

25. sup fig 3b-e – sham operated mice? The best adequate control would be the AAV control DREADD – authors should explain the reason to use sham operated animals.
26. Sup table 4 legend typo
27. Page 13: “Two overlapping clusters were observed, allowing a general separation of the addicted from non-addicted mice” – sup. Fig 5b – this is not clear to me, and no stats is provided to support this claim.
28. Sup fig 5b: ok to separate between addicted and non-addicted but color code WT and CB1 animals in the PCA analysis
29. Sup fig 7 legend is incomplete, not mentioned what data are from controls vs DR2 overexp.
30. Sup fig 9: in the PL terminals in the NAc, authors present CB1 on the resilient phenotype but no D2R and D2R in vulnerable but no CB1 – this is not accurate as CB1 is expressed also in terminals of vulnerable mice right? (and so on)
31. Sup. Fig 9: typo in dopaminergic neuron (appears neuron)

Additional Suggestions:

1. Manuscript should be read by an English-native speaker. Some sentences are awkward, some grammatical errors.
2. Consistently authors drive hypothesis based on very sparse data (eg. Last sentence of page 9), I would refrain of doing this. Be more to the point, less inferences, the data speaks per se.
3. I usually do not interfere much with discussion – as it is a space for authors to have “more freedom” in their interpretations, but I would refrain from allusions to “therapeutic research possibilities” – page 22. I would “tone down” the conclusion section.

I hope that my remarks help improve the manuscript. I truly think that this manuscript has a lot of potential, but it is missing some important control experiments to be publishable. Good luck with your experiments!

Reviewers' comments:

Reviewer #1 (Remarks to the Author):

This study explores the pathways and mechanisms that drive addictive/compulsive behaviors towards highly palatable food. The study identifies the prelimbic (PL) cortical area of mice and their projections to the nucleus accumbens as important in exerting inhibitory control over food intake in regards to the development of perseverative responding, motivation and punishment resistant taking. Using cell specific knockout mice for CB1R and manipulations of neuronal excitability using chemogenetic and Drd2 overexpression in the PL, the study shows that strengthening of the PL-NAC pathway promotes resilience while decreasing the excitability of the PL-NAC neurons generates vulnerability for compulsive taking of highly palatable food.

This is a rational, thoughtful and well-done study. It is clearly written and the logic and rational of experiments is easy to follow. Most important methodological information is included and the conclusions reached are logical and not overstated for the most part (but see below). Appropriate controls are performed and shown (but see one more suggested).

I have a few comments and a suggestion for a control that I think that can improve the clarity and the extend the relevance by highlighting the selectivity of the approach.

- We thank the reviewer for her thoughtful comments. We have answered all her questions and performed the new experiments proposed.

Main comments:

1. There is an important control for the loss of CB1R modulation of glutamatergic transmission in the cortex of the Glu-CB1 KO mice, which targets telencephalic glutamatergic neurons. This is shown in Figure 2 and suppl. Fig 2 too. Now, inclusion of a control for the selectivity of the manipulation would be also very important. For example, is there still modulation of GABAergic transmission in the cortex of these KO mice. Knowing this would allow the authors to further pin point the behavioral effects as being cause by that loss of glutamatergic transmission modulation, which to my view, it is implied there but not really shown (see other two points below as well).

- Following the suggestion of the reviewer, we evaluated the selectivity of the manipulation by analyzing GABAergic transmission in the medial prefrontal cortex (mPFC) and nucleus accumbens (NAc) of Glu-CB1-KO and WT mice. Results showed that there were no differences in the miniature inhibitory postsynaptic currents (mIPSCs)

frequency between genotypes. Thus, there are no differences in GABAergic transmission between both genotypes. Therefore, the behavioral phenotype described in the mutants can be now explained by the specific modulation of glutamatergic transmission that does not affect GABAergic transmission.

- Furthermore, we performed an additional experiment of electrophysiology *ex vivo* in NAc by applying the CB₁R agonist WIN55,212-2 in mutants and WT mice. With this experiment, we now validated the functional relevance of CB₁R signaling at glutamatergic presynaptic terminals by modulation of synaptic glutamatergic release. As it was expected, the EPSCs amplitude was strongly reduced in WT mice after WIN55,212-2 application compared to the baseline, which indicates a functional activation of CB₁R signaling. In Glu-CB1-KO, EPSCs amplitude was still slightly diminished by WIN55,212-2 application, but significantly lower than in WT mice. This partial effect of WIN55,212-2 in mutants could be explained by the expression of CB₁R in other non glutamatergic terminals projecting into the NAc.

Overall, we demonstrated that the absence of CB₁R in glutamatergic terminals in the NAc of Glu-CB1-KO mice had a modulatory effect of the excitatory glutamatergic transmission in the NAc, which contributed to the enhanced inhibitory control of food operant seeking behavior observed in the Glu-CB1-KO mice.

- We provided the additional figures, methods and discussion of these new findings in the manuscript in the corresponding sections. The new figures are represented below and correspond now to the Fig 2a-l and Fig.2 s-t:

-

Fig. 2. Increased synaptic excitatory transmission of glutamatergic PL mPFC and NAc neurons in Glu-CB1-KO. **a, h**, Schematic drawings and infrared differential interference contrast images showing the recorded neuron in the **a**, PL and **h**, NAc. Scale bars, 30 μ m. **b, e** Representative traces of miniature excitatory and inhibitory postsynaptic currents (mEPSCs, mIPSCs) recorded at a holding potential of -70 mV in PL and **i, l**, -75 mV in NAc from WT (black traces) and Glu-CB1-KO (blue traces) mice. **c, d**, mEPSCs frequency and amplitude and **f, g** mIPSCs frequency and amplitude of WT (black) and Glu-CB1-KO (blue) mice in the PL and **j, k, m, n** in the NAc (mean \pm S.E.M; t-test * $P < 0.05$, ** $P < 0.01$, 9-12 cells from 12 mice;). **o**, Paired pulse facilitation (PPF). Representative overlapped recordings of L5 field excitatory postsynaptic potential (fEPSP) before (P1) and after (P2) stimulating twice in layer 2/3 with an interpulse interval of 50 ms for WT (above) and Glu-CB1-KO (below). **p**, Increased paired pulse facilitation ratio (P2/P1) in Glu-CB1-KO compared to WT (U Mann-Whitney ** $P < 0.01$; 12-14 slices from 5 animals per genotype). **q**, Representative overlapped recordings of the modulation of PSPs amplitude before and after application of the CB1R agonist WIN55,212-2 (5 μ M) compared to basal conditions in PL for WT

(above) and Glu-CB1-KO (below). **r**, Changes of PSPs amplitude in PL for WT and Glu-CB1-KO (paired t-test *** $P < 0.001$; 6 slices from 3 animals per genotype). **s**, Representative synaptic responses showing mean EPSCs (5 consecutive EPSCs) before and after CB1R agonist WIN55,212-2 (5 μM) bath application in the NAc for WT (above) and Glu-CB1-KO (below). **t**, Average relative changes of EPSCs amplitude in NAc of WT and Glu-CB1-KO mice (paired t-test * $P < 0.05$, ** $P < 0.01$; 7-10 cells from 4 animals per genotype; mean and individual values; see also Supplementary Fig. 3; statistical details are included in Supplementary Table 2).

- In the results and discussion sections, we included the following paragraphs:
“We found that mEPSCs frequency was increased in Glu-CB1-KO compared to WT mice, suggesting an enhanced probability of glutamate vesicle release onto pyramidal L5 PL neurons that was independent of presynaptic voltage Ca^{2+} ion channels ($P < 0.01$, Fig. 2a-d). Notably, no significant differences were found in the miniature inhibitory postsynaptic currents (mIPSCs) frequency and in amplitude between Glu-CB1-KO and WT mice (Fig. 2e-g). In a next step, we evaluated if the glutamatergic and GABAergic synaptic transmission in NAc was also altered by the loss of CB₁R in cortical glutamatergic cells. Electrophysiological recordings in the NAc showed an increased mEPSCs frequency and no differences in the mIPSCs in Glu-CB1-KO mice compared to WT mice ($P < 0.05$, Fig. 2h-k). Thus, the lack of CB₁R in dorsal telencephalic glutamatergic neurons increased synaptic excitatory transmission in PL cortex and NAc without affecting inhibitory synaptic transmission in these areas.” (results section, page 8, line 174-185)
- *“Similar results were obtained evaluating the changes of EPSCs amplitude in the NAc. The EPSCs amplitude was strongly reduced in the WT mice after WIN55,212-2 application compared to the baseline. In Glu-CB1-KO mice, EPSCs amplitude was still slightly diminished by the WIN55,212-2 application, but in a significantly lower extension than in WT mice ($P < 0.01$, Fig. 2s-t). This partial effect of WIN55,212-2 in the mutants could be explained by the expression of CB₁R in other non-glutamatergic cell-types or projections in the NAc.”* (results section, page 10, line 203-209).
- *“This resilient phenotype was mediated by increased glutamatergic transmission in mPFC and in the NAc. Specifically, our electrophysiological recordings of mEPSCs frequency in Glu-CB1-KO mice revealed an enhanced excitatory synaptic transmission onto pyramidal glutamatergic neurons in L5 of the PL cortex and in their targeted MSNs in the NAc. This effect was specific of glutamatergic transmission since the GABAergic synaptic transmission in these mutants was not altered. Additionally, the PPF results revealed that the deletion of presynaptic excitatory CB₁R produced an enhancement in glutamate vesicle release in the PL cortex that is action potential dependent. Altogether,*

these results highlight the involvement of the glutamatergic PL-NAc projections in the behavioral phenotype of inhibitory control observed in Glu-CB1-KO mice.” (discussion section, page 19, line 424-434).

- *“Furthermore, an additional experiment of electrophysiology ex vivo in the mPFC and NAc with the application of the CB₁R agonist WIN55,212-2 in Glu-CB1-KO and in the WT mice was performed. With this experiment, we validated at functional level the deletion of CB₁R signaling at glutamatergic presynaptic terminals in both areas of Glu-CB1-KO mice. As it was expected, the amplitude of fPSPs in the mPFC and the amplitude of EPSCs in the NAc were strongly reduced in WT mice after WIN55,212-2 application compared to the baseline, which indicates a functional activation of CB₁R signaling. This effect was blunted in the conditional mutant mice, confirming the deletion of the CB₁R in glutamatergic cells in mPFC. On the other, in the NAc, the EPSCs amplitude was still slightly decreased after WIN55,212-2 application in Glu-CB1-KO mice, but significantly lower than in WT mice possibly due to the presence of CB₁R in other non-excitatory cell-types (Marsicano and Kummer 2008; Robbe et al. 2001).” (discussion section, page 20, line 442-452).*

2. The text seems to put too much emphasis on the glutamatergic transmission in the cortex. It is true that this is one of the factors altered in the CB1R KO mice but in general the study seems to show that it is actually the overall activity and the engagement of the L5 pyramidal neurons in the PL cortex and their projections to the accumbens which are important on controlling addictive behaviors towards palatable food. So the study will benefit from revising the text and the discussion to represent this.

- According to this suggestion, we discussed more in detail the involvement of the PL-NAc network in the Glu-CB1-KO mice. The following sentences have been included in the discussion section:

“This resilient phenotype was mediated by increased glutamatergic neuronal activity in mPFC and in their projections to the NAc.” (page 19, lines 424-425)

- *“This top-down mechanism involving PL-NAc projections seems crucial in the resilient phenotype of the mutants and was further investigated in our study. Importantly, addiction-like behavior is associated with an alteration in the synaptic excitatory transmission of prefrontal cortical areas that normally provide inhibitory control over striatal activity (Wang et al. 2012).” (page 21, lines 546-460)*

3. There are several occasions in which the cells losing CB1R and the cells recorded and showing the effect are being confused or presented in a very confusing matter. This is especially true in the discussion section (page 19), where the pyramidal neurons lacking CB1R are recorded but the changes measured are unlikely due to the loss of the CB1R on these cells but rather on the presynaptic glutamatergic neurons that form synapses onto them and where CB1R presynaptically inhibit glutamate release.

- We modified the following text in the discussion section accordingly to this helpful thought (page 19, lines 425-428):

“Specifically, our electrophysiological recordings of mEPSC frequency in Glu-CB1-KO mice revealed an enhanced excitatory synaptic transmission onto pyramidal glutamatergic neurons in L5 of the PL cortex and in their targeted MSNs in the NAc.”

4. What is the body weight of Glu-CB1 KO mice and how it compares to WT? how could the body weight information affect the interpretation of the findings with regards to lower motivation and compulsivity?

- In agreement with the reviewer's comment, we now highlighted this issue in the results and discussion sections. The body weight variable does not seem to be relevant for the interpretation of the lower persistence to response, motivation and compulsivity shown by Glu-CB1-KO mice. To evaluate the possible influence of body weight in the behavioral response, we measured the progression of body weight during the whole experimental sequence of 24 weeks, the total average of body weight separately in the early and late period, and the possible correlations between the body weight and the three addiction criteria depending on the genotype. Interestingly, no significant correlation between the body weight variable and each of the three addiction criteria was revealed in both early and late periods (Supplementary Fig. 2c-e).
- Glu-CB1-KO mice showed significantly lower body weight than WT mice when the progression of body weight was measured weekly during the whole experimental sequence of 24 weeks (Supplementary Fig. 2a, repeated measures ANOVA, genotype effect $^{**}P < 0.01$). However, no differences in body weight between genotypes were revealed in the early period of the food addiction protocol, where Glu-CB1-KO mice already showed significantly less compulsivity than WT mice (Supplementary Fig. 2b). This result was in accordance with previous studies showing no alteration in body weight when mice were fed *ad libitum* with regular chow in a short period of time (Bellocchio et al. 2010). In the late period, the body weight differences between genotypes emerged, although the correlations between the body weight variable and the three

addiction criteria were not significant (Supplementary Fig. 2c-e). Therefore, the body weight variable is not a confounding factor.

- To highlight this issue, we included the following two graphs representing the body weight across weeks and the body weight in two time points, the early period and the late period depending on the genotype (Supplementary Fig. 2a-e). These findings are mentioned in the results section (page 7, line 151-162) as follows:

“To study if the resilient phenotype of Glu-CB1-KO mice was influenced by the body weight variable, we measured the evolution of the body weight during the whole experimental sequence of 24 weeks, the total average of body weight separately in the early and in late period and the correlations between the body weight and the 3 addiction criteria depending on the genotype (Supplementary Fig. 2a-e). The lack of significant differences between genotypes in the body weight in the early period, when mutants showed significant increased compulsivity, suggested that the body weight is not a predisposing factor in the development of the addictive-like behavior. In the late period, body weight differences between genotypes emerged, although the correlations between body weight and the 3 addiction criteria were not significant indicating that the body weight variable does not explain the lower persistence to response, motivation and compulsivity found in the Glu-CB1-KO mice.”

- These findings are mentioned in the discussion section (page 19, line 416-421) as follows:

“Notably, Glu-CB1-KO mice did not show changes in body weight in the early period similarly to what was previously reported when mice were fed ad libitum with regular chow in a 12 weeks period of time (Bellocchio et al. 2010). In the late period, mutants showed lower body weight than WT mice but this variable did not affect the resilient phenotype since no correlations between the 3 addiction criteria and body weight were revealed”

Supplementary Fig. 2.a-e, Body weight. **a**, Weekly measures of body weight in grams (Repeated measures ANOVA, Genotype effect $**P < 0.01$). **b**, Mean body weight of the early period and late period (t-test, $**P < 0.01$). **c-e**, Correlation between the body weight (g) and the 3 addiction criteria in both genotypes. **c**, Persistence to response. **d**, Motivation. **e**, Compulsivity ($n = 56$ for WT mice and $n = 58$ for Glu-CB1-KO mice; statistical details are included in Supplementary Table 1).

5. It would help to sharpen the terminology of the ephys mini data. The results section refers to “spontaneous” release but these are miniature responses obtained in TTX, then the authors could skip the spontaneous term and also maybe make inference on release probability.

- In agreement with the reviewer's comment, we deleted the word “spontaneous” from the results section. We also made reference on release probability by introducing the following phrase (page 8, lines 174-177):

“We found that mEPSCs frequency was increased in Glu-CB1-KO compared to WT mice, suggesting an enhanced probability of glutamate vesicle release onto pyramidal L5 PL neurons that could be independent of presynaptic voltage Ca^{2+} ion channels ($P < 0.01$, Fig. 2a-d).”

Minor comments:

6. Page 17, last sentence of the result section is a bit confusing or inaccurate. In the sense that the D2R do not decrease the excitability but rather dopamine via D2rs. It could be revised.

- We modified the following sentence accordingly (page 18, lines 390-392):
“In summary, we revealed that overexpression of D₂R allowed that dopamine via D₂Rs decreased the excitability of PL-NAc core projections, conveying the vulnerability to develop food addiction.”

7. Figure 6, panel A. Wrong labels. They are the same as figure 5 and should be fixed to reflect the Drd2 overexpression manipulation.

- We corrected this mistake in the Fig. 6a (currently Fig. 5a).

8. DREADD figures. Very nice that study includes and shows the controls showing the manipulations work as expected. Could the authors move to main figure the control of CNO application in the absence of the hM4Di expressing to the main figure. At least for the ephys parameter shown in the figure. (others could stay in the suppl. Figure).

- As suggested by the reviewer, the figures of CNO application in the absence of the hM4Di-expressing cells and D2R-expressing cells has no effect in the firing rate have been moved to the main figures (now Fig. 3d and Fig. 5d).

9. Page 18- “synaptic transmission of pyramidal glutamatergic neurons” might be more clear if it reads “on” or “onto”

- We included the following new sentence (page 19, lines 425-428):
“Specifically, our electrophysiological recordings of mEPSCs frequency in Glu-CB1-KO mice revealed an enhanced excitatory synaptic transmission onto pyramidal glutamatergic neurons in L5 of the PL cortex and in their targeted MSNs in the NAc.”

10. Page 19. “decrease selectively the glutamatergic activity of the PL region” is confusing.

- Accordingly, we changed the sentence as follows (page 21, lines 465-467):
“Therefore, we decreased the glutamatergic transmission or excitatory neuronal activity of the PL region by using a Cre-dependent inhibitory chemogenetic approach in Nex-Cre mice.”

11. Personally, statements such as “for the first time...” do not add much.

- We deleted the statement “For the first time”.

Reviewer #2 (Remarks to the Author):

Domingo-Rodriguez and collaborators report an impressive set of results indicating that CB1 cannabinoid receptors expressed by glutamatergic neurons in the median prefrontal cortex regulate the development of addictive-like behaviors in a mouse model of 'food addiction'. They further show that this response is at least partly mediated by mPFC projections to the nucleus accumbens, and that the development of 'food addiction' is accompanied by profound changes in the expression of dopamine D2 receptors in the mPFC. The experiments are conducted with the expertise and quality that is the norm for these excellent investigators. The manuscript is clear and well written. The interpretation of the data is sound, though at time hyped and incomplete (as explained below).

- We thank the review for her/his positive reception of our work. We have included the changes suggested by the reviewer and we have performed the new experiments proposed. For clarity, we specify our answers point by point as it is explained below.

1. A first conceptual issue is the apparently a critical acceptance of food addiction as a construct. I am agnostic on this point but given the present controversy I recommend that the authors do a better job at positioning their study. The statement that 25% of the world population is affected by food addiction, taken from a review, is implausible.

- We followed this critical point by introducing the controversy of the concept of food addiction in the introduction section as follows (page 3, line 47-52):

“The concept of food addiction is still controversial being also under debate if it could represent a behavioral addiction or whether specific components of food could have intrinsic addictive properties similar to drugs of abuse (Gordon et al. 2018). However, the

diagnosis of food addiction is not included in the 5th edition of the Statistical Manual of Mental Disorders (DSM-5), although a validated tool, the Yale Food Addiction Scale (YFAS) is widely accepted among the scientific community (Gearhardt, Corbin, and Brownell 2016).

- We removed the statement that 19.9% of the world population is affected by food addiction.

2. Another conceptual issue pertains to the mouse model used in the study. Previous work has shown that removal of CB1R from telencephalic glutamatergic neurons has a marked impact on food intake. What is the possible contribution of this trait to the changes observed? A deeper discussion is needed about this point.

- We modified the discussion to incorporate this important point. The contribution of this trait to the changes obtained in the food addiction model could represent a protective factor leading to a decrease in the reinforcing effects of chocolate flavored pellets over the time. To reflect this point in the manuscript, we introduced the following paragraphs in the results and discussion sections:

- *“In FR1, both genotypes increased the number of reinforcers across sessions without significant differences indicating similar levels of acquisition of the operant conditioning learning. However, when the effort to obtain one single pellet was increased to FR5, the progressive increase of the number of reinforcers was significantly reduced in Glu-CB1-KO as compared to WT (genotype effect, $P < 0.001$; interaction genotype \times sessions, $P < 0.001$, Fig. 1b) leading to a reduced number of reinforcers over the entire FR5 period. This result suggests that palatable pellets were less reinforcing for the mutants since the FR5 early period ($P < 0.01$, Supplementary Fig. 1a) and this trait may represent an initial protective factor.”* (page 5, lines 110-118).

“Previous studies reported that the deletion of CB₁R from dorsal telencephalic glutamatergic neurons has a marked impact on food intake in fasting conditions (Bellochio et al., 2010). However, these mice did not showed body weight changes when were fed ad libitum (Bellochio et al., 2010). Our results showing decreased palatable pellets intake by Glu-CB1-KO mice without food restriction are consistent ($P < 0.001$, Fig. 1b).” (page 7, line 147-151).

- *“Moreover, a phenotype with a reduction of the exploratory behavior (Häring et al. 2011), increased neophobia, high passive fear response after conditioning, and decreased food intake after fasting was described in these mutants (Bellochio et al.*

2010; Lutz et al. 2015). *These previous studies reported an effect on food intake when CB1R was deleted from dorsal telencephalic glutamatergic neurons but exclusively after a 24 h fasting period.. Our results suggest a decreased reinforcing effects of chocolate-flavored pellets in the mutants, which could contribute to the protective phenotype to develop the multifactorial food addiction disorder.*” (page 19, lines 409-416)

3. I would recommend increasing the number of samples in the physiological experiment of Figure 2b. As it stands, the claimed difference can be ascribed to only two samples and is not credible.

- Following this suggestion of the reviewer, we increased the number of samples in the paired pulse facilitation experiment (Fig. 2p). In the new version, we confirmed our previous data by including a total of 12-14 slices from 5 animals per genotype.

4. The transcriptome analyses indicate a large elevation in Drdr2 in 'addicted' versus 'non-addicted' mice. This is really not a great surprise, given the role of this receptor in motivated behavior. But the analyses also show an elevation in Gpr88, which is a lot more intriguing and potentially novel. Incidentally, the names of genes whose expression is reduced should also be listed and discussed.

- We agree that the D₂R expression has been strongly linked with motivated behaviors and a down-regulation of D₂Rs has been found in the striatum of cocaine addicted and obese individuals (Volkow et al. 1993; Wang et al. 2001). The novelty of our study is that food addicted mice showed an up-regulation of *Drd2* gene in a different brain area, the mPFC. We hypothesized that this upregulation is modulating the excitability of the glutamatergic cortico-striatal projections (PL-NAc core) promoting the addictive-like behavior. We highlighted the relevance of this new finding in the discussion section as follows (page 24, line 526-533):

- *“Notably, the most differentially expressed gene found in our study was the *Drd2*. To our knowledge, this is the first study revealing an increased expression of the gene encoding for *D₂R* in the mPFC in the context of addiction. In contrast, a reduced levels of *D₂R* in the striatum have been already implicated in this disorder (Volkow et al. 1993). Thus, neuroimaging studies reported a downregulation of *D₂R* in the striatum, which correlated with an hypofunction of the PFC in cocaine abusers (Volkow et al. 1993). In agreement, a specific *Drd2* polymorphism is associated with the “Reward Deficiency Syndrome”, consisting in a hypodopaminergic state due to the altered *D₂R* function (Blum et al. 1996).”*
- We included new sentences in the discussion section to address the possible relevance of the upregulation of the *Gpr88* gene in food addicted animals (page 23, line 514-517):
- *“Interestingly, inactivation of *Gpr88* gene enhanced the excitability of both *D1* and *D2* MSNs in the striatum (Quintana et al. 2012). Therefore, we could speculate that upregulation of *Gpr88* in the mPFC of addicted mice could also contribute to the alteration of cortico-limbic system in these mice.”*
- As suggested by the reviewer, we added in the current version of the manuscript 4 detailed tables listing all the genes upregulated and downregulated comparing addicted and non-addicted mice and comparing WT and Glu-CB1-KO mice (Supplementary Table7-10). Moreover, the following paragraphs were added to the results and discussion sections as follows:
- *“Several genes such as *Myh11* (myosin heavy chain 11), *Acta2* (actin alpha 2), *Cdh1* (cadherin 1), *Ptgds* (prostaglandin D2 synthase), and *Fosb* (fosb proto-oncogene, AP-1 transcription factor subunit) were downregulated, suggesting changes in neuronal plasticity, prostaglandin synthesis and gene regulation” (results section, page 14, line 307-310).*
- *“*Dnah6*, *Spaca1* and some small nucleolar genes (*Snora68*, 78, 70, 8, 31) were upregulated.” (results section, page 15, line 320-321).*
- *“In addition, several genes in the mPFC were downregulated in addicted mice independently of the genotype. Interestingly, genes (*Myh11*, *Acta2* and *Cdh1*) related to neuronal plasticity (Kneussel and Wagner 2013; Tan et al. 2010), prostaglandin synthesis (*Ptgds*) (Urade and Hayaishi 2000) and gene regulation (*Fosb*) (Nestler 2013) were differentially expressed. The transcriptomic analysis between WT and Glu-CB1-KO mice revealed a downregulation in the *Cnr1* gene as expected in the mPFC of the mutants confirming the genetic deletion of the *CB₁R* in this area. Additionally, *Fos* gene, encoding for c-fos protein, was less expressed in the mutants implicating an enhanced neuronal*

activity in mPFC of WT compared to Glu-CB1-KO mice". (discussion section, page 24, line 534-541)

5. I would be careful in the use of the word 'addicted' etc. A key component of addiction is the persistence in a behavior that is recognized as detrimental. There is no such recognition in the mouse. Phrases like 'addiction-like' etc appear to be more appropriate in this context.

- We changed the word addicted or addiction for the most correct addiction-like.

Reviewer #3 (Remarks to the Author):

In this manuscript, authors elucidate some of the mechanisms underlying food addiction "vulnerability" by using KO animals, chemogenetics, gene overexpression experiments and performing behavioural, electrophysiological and RNA seq analysis.

Authors show that CB1 deletion in dorsal telencephalic glutamatergic neurons changes food addiction profile. They then show that mPFC-NAC core projections silencing by DREADDs increases compulsive food seeking. RNAseq revealed that D2R was upregulated in the mPFC of addicted animals, so authors overexpressed D2R in this brain region, which promoted food addiction phenotype.

Some of the results of the manuscript are very interesting and novel, and experimental design was performed adequately and with state of the art tools. However, I think that there some limitations that require attention. Please see my comments below, I hope it helps authors to improve the manuscript and strengthen their conclusions.

- We thank the reviewer for the positive comments and very useful suggestions. Below it can be found the answers to all the comments point by point.

As a general remark, I missed coherence and integration of the different parts of the manuscript. In the beginning that is not completely clear why authors include the CB1-KO data – I think this data is really nice, but in my opinion, it does not add much to the other part of the story regarding D2R (and vice versa) (and similar CB1-KO data was previously published by the team – NPP 2015). The two parts lack integration (authors also appear to partially agree with this, as they state in page 15 "we hypothesized that this (D2R) upregulation could play a key role in the development of food addiction behavior, irrespective of the presence or absence of CB1R."

- After the deep revision of the manuscript, we improved the message of the role of CB₁R in glutamatergic cells compared to previous studies and better integrated these experiments that complement those performed to evaluate the role of the D₂R in the PL-NAc projections. To integrate these two parts, we performed all the crucial electrophysiological experiments suggested by the reviewer. Thus, electrophysiological recordings into the NAc in Glu-CB1-KO mice and in mice overexpressing D2Rs are now included in the revised version of the manuscript (see the answer to the major comment, 7, 8, 9 and 10 and see Fig. 2, 3 and 5).

Yes, both endocannabinoid and dopaminergic control of PL neurons modulate food addiction, but how do they interact (if they do)? If it is just a matter of modulation of PL-NAc glutamatergic signalling (as I think authors postulate), then, for example, you could register the activity of striatal D2-MSNs (and D1-MSNs) in glu-CB1KO to show that glutamatergic signalling is indeed increased in the NAc NO GO pathway, as authors hypothesize. Then block it to see if the resilient phenotype is abolished?

- We agree that our findings revealed that the endocannabinoid and dopaminergic systems control the PL-NAc neurons to modulate the food addiction phenotype. As suggested by the reviewer we have now registered in the NAc the activity of the MSNs in Glu-CB1-KO mice and in mice overexpressing D₂R. The new results show that the synaptic glutamatergic transmission was increased onto MSNs in the NAc in Glu-CB1-KO mice while was reduced in D₂R-overexpressing mice. The reduced synaptic glutamatergic transmission from PL-NAc pathway in D2R-overexpressing mice is conferring a susceptibility phenotype to develop food addiction. A similar observation was found in WT animals using an inhibitory chemogenetic approach of PL-NAc projections promoting a vulnerable phenotype (Fig. 3). Therefore, our present data demonstrate that this specific PL-NAc pathway is involved in the development of food addiction.
- In addition, we included new experiments of electrophysiology ex vivo to evaluate the excitatory and the inhibitory transmission (mEPSCs and mIPSCs) in mPFC and NAc of Glu-CB1-KO mice. The new results show that there were no differences in the mIPSCs frequency between Glu-CB1-KO and WT mice. Thus, the modulation of GABAergic transmission in Glu-CB1-KO mice was not altered in these two brain areas. Therefore, the behavioral phenotype described in the mutants can be now explained by the specific modulation of glutamatergic transmission that does not affect GABAergic transmission.

- The details of the experiment of electrophysiology in the NAc of the Glu-CB1-KO mice are explained in the major question 10 that refers to the Fig. 2 of the current manuscript.

One other caveat of the study is that ephys recordings were always made in the L5 and never in the NAc, so it is risky to assume that the resilient and vulnerable phenotype arises from this particular connection, as authors assume. Both CB1-KO and D2R overexpression experiments of this study do not exclude the contribution of other brain regions other than NAc for the phenotype. For example, glu-CB1KO can have effects both at L5 level or presynaptically at PL terminals in the NAc core; the same is true for D2R overexpression, which can occur both at PL/L5 levels or terminals in the NAc. Some degree of PFC-NAc collateralization exists (Pinto, Sesack 2000), so this should be taken into consideration in this study and future experimental designs.

- To assume that the resilient and vulnerable phenotype in the Glu-CB1-KO and D2R overexpressing mice arises from this particular PL-NAc core connection, respectively, we registered the excitatory transmission onto the MSNs in the NAc. The recordings in the NAc of WT showed a strong reduction in the EPSCs amplitude after CB₁ agonist (WIN55,212-2) application demonstrating the functional relevance of presynaptic CB₁R in the glutamatergic terminals. On the other, the recordings in the NAc in mice overexpressing D2R revealed a decrease in the EPSCs amplitude after the D2R agonist quinpirole. These new results with our previous data of the reduction of firing rate after quinpirole application in the mPFC suggest that the overexpression of D2R in glutamatergic neurons has an effect in both areas, the mPFC and the NAc. This effect can be mediated by D₂R in postsynaptic dendrites and/or presynaptic terminals. Together, these new experiments provided new insights to the study and highlighted the role of the glutamatergic transmission from the PL to the NAc in the development of the food addiction-like behavior.

Using the same strategy as in this manuscript, if you overexpress D2R in PL-Amyg neurons for example (or other PL-connection), would you see the same effect in food addiction or not? (this could help control if this effect is exclusive to PL-NAc core terminal overexpression or is more related to overexpression of D2RD in PL from example). Similarly, if you knockdown D2R in PL-NAc neurons, what is the effect in the development of food addiction? The same control experiment could be performed for DREADD experiment in order to show that it is PL-NAc projections that are crucial for food addiction (and not other PL- projections). I believe these experiments would be important to disentangle the “specificity of projections” question.

- In our experiments, we revealed that the modulation of the mPFC projections to NAc is sufficient to produce the endophenotype of compulsivity to palatable food seeking. Certainly, the complex phenotype of addiction could also be influenced by the activity of several other pathways that could be elucidated in other research projects. The experiment targeting other networks such as the PL-amygdala represents a different research project to evaluate novel potential pathways that could also be involved in addiction. The additional projects proposed by the reviewer to knockdown D₂R in PL-NAc neurons or about other related networks could also be of potential future interest. However, to evaluate the consequences of this manipulation that would be potentially protective for the development of compulsivity and food addiction, we would require a long period of 118 days of operant conditioning behavior. Thus, this experiment would require at least 6 months to obtain the results and are beyond the scope of the present study.
- The performance of the initial experiments proposed by this reviewer (D₂R downregulation in PL-NAc and PL-Amyg plus D₂R upregulation in PL-Amyg) would represent at least 18 months of work. If the same experiments are performed in the DREADD approach, as suggested by the reviewer, we would require a total of at least 36 months (18+18 months) just to obtain the results to answer this specific reviewer's comment. In agreement with this reviewer's comment, we now acknowledged in the discussion that our results open new studies to elucidate other possible neuronal circuits that could also participate in the development of food addiction (page 22, line 497-498). However, these new experiments would require several years of work before obtaining conclusive results.

Also, transcriptomic analysis is very incomplete; full data is not provided. Authors also do not show evidence of D₂R overexpression.

- These two last statements are answered in detail in the specific questions below, major question 6, fig5.a-d and major question 7, Fig. 6b-d.

Major

1. **Statistics: Initial glu-CB1KO data is outstanding, with a large number of animals, but for some other experiments, authors did not provide G power analysis (or similar) to have a sufficient/adequate number of animals. Though one can justify that we can use "usual" number of animals as previous studies, in this work, authors are dealing with a small**

percentage of animals that will develop food addiction, so these calculations should be taken into consideration (see below some of my comments).

- In the initial experiment using Glu-CB1-KO mice in the food addiction protocol, we used a higher number of animals than the previous study using the same experimental sequence (Mancino et al., 2015). In this earlier study, the initial population was genetically heterogeneous and required a sample size of n=11-30 per group to perform a genetic selection of addicted and non-addicted mice based on the phenotype. In our study, mice were inbred, and the genetic variable was homogenous. Therefore, a larger number of mice (n=56-58 per genotype) than in the previous was required to obtain such a variability in a inbred homogeneous strain. In turn, the experiments with the viral vector approach required a similar sample size than previous studies (n=12-22 per group) because we were doing a genetic manipulation to induce a vulnerable phenotype using the short food addiction protocol. Indeed, we have performed a power analysis under these experimental conditions and consequently, we have increased the number of mice of the experiment of chemogenetic inhibition of the PL neurons from n=8 to n=14. This range of animals is within the range calculated by our power analysis. Indeed, 12-22 mice per group of our experiments with genetic manipulation and the two-sample t-test achieved a power analysis between a range of 73-90% ($\alpha = 0.05$). Thus, our power analysis has revealed that the current number of animals per group in all the experiments is adequate to find a statistical significance. To reflect this statement, we have introduced the following sentence in the Supplementary materials section (page 21, line 479-482).

“The sample size was calculated based on the power analysis. The criterion for significance (α) was set at 0.050 and the statistical test used was two-sample t-test. With the sample size of 12-22 mice per group, our studies achieved a power between 73-90%”.

2. Fig 1b – shows infection of ACC also – this should be discussed.

- We agree that the expression was localized in the PL area and in some mice the expression affected also a part of the ACC. We acknowledged this point in the discussion section as follows (page 21, line 470-473):
- *“We cannot discard the possible involvement of the anterior cingulate cortex in this behavior, an area which is anatomically grouped with the PL region to the dorsal mPFC*

(Heidbreder and Groenewegen 2003), that corresponds to the dorsolateral PFC in humans (Vertes 2006), an area classically involved in response inhibition (Moorman et al. 2015)”

3. Fig. 1b vs 3e- number of reinforcers in 1h vs 2h, respectively- why do authors do not use the same period of time?

- We increased the time of the operant sessions in the chemogenetically and gene overexpression experiments to potentiate the emergence of the vulnerable phenotype in a short period of time. Using this procedure, we increased the exposure of the palatable pellets each day to ensure the development of the addiction-like phenotype. To clarify this question, we added the following sentence in the supplementary materials (page 3, line 64-68):

“Daily self-administration sessions maintained by chocolate-flavored pellets lasted 1 h in the long food addiction protocol and 2 h in the short food addiction protocol to increase the exposure of the palatable pellets each day to ensure the development of the addictive-like phenotype.”

4. Fig. 3: this figure presents some errors and needs some clarifications.

a. In which layer of PL did you perform the recordings of Fig. 3c? (I assume it is L5 but this should be referred in results section).

- The recordings of PL in the experiment of the Fig. 3c were performed in L5. This point has been clarified in the supplementary results section (page 24, line 510-513).

“Next, we aimed at validating our approach by using whole-cell current clamp recordings in L5 of visually identified hM4Di-mCherry expressing PL neurons in the presence of the selective ligand clozapine-N-oxide (CNO).”

b. Authors should also explain the heterogeneous firing rates of PL neurons – this is not usual. Any explanation?

As the reviewer pointed out, we revealed certain heterogeneity of the firing rates of PL neurons. Accordingly, it has been previously described that the neocortex is a highly heterogeneous region, in the population of both interneurons and pyramidal neurons, even within the same layer. In detail, a pyramidal diversity is described in the different layers of medial prefrontal cortex (including layer 5) of rat (Van Aerde and Feldmeyer 2015). Therefore, we think that the heterogeneity found in this study could suggest that different subtypes of pyramidal neurons project to the NAc.

- c. There is represented only one addicted mice in Fig3f in saline group, then no addicted mice appear in 3g-h in the same saline group? Explain.

We have now corrected the mistake raised by the reviewer in one labeled point of Fig. 3g-h (currently moved to Supplementary Fig.4g-i).

- d. Also regarding 3i: if there is only 1/7 addicted animal in saline group, how is the percentage 12.0%? How was this calculated?

- The percentage of the addicted mice in the saline group represented in the Fig. 3i was 12.5%. It was calculated considering 1 addicted mouse out of 8 total mice (1/8). As it was shown in the figures 3f-h and mentioned in the results section, the total number of mice in this group was n=8. In the current version of the manuscript, we have increased the final number of animals per group to n=13 in saline treated mice and n=14 in CNO treated mice to answer the reviewer's comments of question 4f. Thus, the new percentage is 15.4% of addicted animals in the saline group compared to 42.8% in the CNO group.

- e. **Small typo: In the table, appears 12.5%, graph appears 12%.**

- The graph Fig.3i (currently moved to Supplementary Fig.4j) was corrected with the new percentage in accordance to the table shown below the figure.

- f. Authors refer that in a large population (Fig. 1), 25% of wt animals will develop food addiction; so using 8-10 animals is clearly insufficient. Was this experiment replicated in another set of animals?

- As the reviewer suggested new animals were added to the presented data to obtain more coherent results. The final number of animals was n=13 in saline and n=14 in CNO treated mice. We obtained a 15.4% of addicted animals in the saline group compared to

42.8% in the CNO group. In the current version of the manuscript, figure 3 (currently moved to supplementary Fig. 4) has been modified including the new group of mice.

- g. Why do authors observe a decrease in persistence in CNO animals? I assume that if you would take the saline addicted animal, the two groups would be identical, which again emphasizes that the number of animals used for this experiment is insufficient.**
- The incorporation of new animals to this experiment, as suggested by the reviewer, minimized this unexpected difference in the persistence to response criterion (Supplementary Fig. 4g).

- h. This figure does not add much to the main conclusions of the manuscript - especially because you have Fig. 4 which is a finer approach, so I would place Fig. 3 in Sup. Material. In addition, you also have substantial transfection of the ACC brain region, which can partially contribute for the observed phenotype.**
- According to the reviewer's comment, the Fig. 3 was moved to supplementary material section. In the current version it corresponds to Supplementary Fig. 4.

5. Fig. 4:

- a. Page 12: "The lack of significant differences in the persistence to response and motivation highlights that the PL-Nac core projection was selectively involved in the loss of control leading to compulsive food seeking" – overstatement, please remove.**
- In agreement to the comment of the reviewer, we deleted the mentioned sentence.
- b. Sup. Fig 4 i-k, l-n : different number of animals in the panels and also does not match main fig 4 – why? Were animals excluded? Based on what criteria?**
- The Supplementary Fig. 4i-k (currently the Fig.3 m-o) show the correlations between the achieved addiction criteria with the value of each addiction criteria. The graphs

represent all individual values introduced, and as the reviewer highlighted, it seems that there were different number of animals in some graphs. This apparent discrepancy is due to an overlapping of the points. Indeed, there were some animals that achieved the same addiction criteria with exactly the same score in the criteria. To clarify this issue, we modified the graphs.

- As the reviewer highlighted, there were two graphs Supplementary Fig. 4m-n (currently supplementary Fig. 6k-m) that were corrected including all mice.

6. Fig 5

- This data is very incomplete; authors do not provide complete list of DE genes; raw data and final analysed data should be provided. This data should also be more integrated and discussed in the paper.**
- Following the reviewer's recommendation, we provided a complete list of all differentially expressed (DE) genes in Supplementary Tables 7-10. Raw data are available using the accession number GE139482 (as indicated in the manuscript). These data are also discussed in results and discussion sections as follows:
 - *"Upon performing differential gene expression analysis between non-addicted and addicted mice, 31 genes were significantly upregulated, whereas 70 genes were downregulated (Fig. 4e and Supplementary Table 7-8). Interestingly, genes previously*

related to the reward system such as *Drd2* (dopamine receptor type 2), *Adora2a* (adenosine receptor 2a), *Gpr88* (orphan G-protein coupled receptor 88), and *Drd1* (dopamine receptor type 1) mRNA were found to be strongly upregulated in the addicted mice. Several genes such as *Myh11* (myosin heavy chain 11), *Acta2* (actin alpha 2), *Cdh1* (cadherin 1), *Ptgds* (prostaglandin D2 synthase), and *Fosb* (*fosb* proto-oncogene, AP-1 transcription factor subunit) were downregulated, suggesting changes in neuronal plasticity, prostaglandin synthesis and gene regulation.” (results section page 14, line 301-310):

- “In addition, several genes in the mPFC were downregulated in addicted mice independently of the genotype. Interestingly, genes (*Myh11*, *Acta2* and *Cdh1*) related to neuronal plasticity (Kneussel and Wagner 2013; Tan et al. 2010), prostaglandin synthesis (*Ptgds*) (Urade and Hayaishi 2000) and gene regulation (*Fosb*) (Nestler 2013) were differentially expressed. The transcriptomic analysis between WT and *Glu-CB1-KO* mice revealed a downregulation in the *Cnr1* gene as expected in the mPFC of the mutants confirming the genetic deletion of the *CB₁R* in this area. Additionally, *Fos* gene, encoding for *c-fos* protein, was less expressed in the mutants implicating an enhanced neuronal activity in mPFC of WT compared to *Glu-CB1-KO* mice”(discussion section, page 24, line 534-541)

b. What was the criteria to show those particular DE genes? Why did authors only selected upregulated genes to be replicated by qPCR in 5e, whereas in 5h were the down-regulated which were selected?

- All genes with a fold change higher than >1.5 fold, p-value < 0.01 and average read counts >40 in either condition comparison (i.e. addicted vs. non-addicted and WT vs KO mice) were selected as differentially expressed genes. Besides this information, we also indicated further DE genes in Fig. 4e and 4g of the revised manuscript. Furthermore, as mentioned above, we provided a complete list of all DE genes in supplemental material (Supplementary Tables 7-10).
- We selected for the qPCR the most relevant genes due to its involvement in addiction and neuronal activity based on previous studies.

c. fig. 5f: use two colors for the dots to distinguish between wt and CB1 KO in the distribution; 5h – color code also between addicted and non-addicted.

- Accordingly to the reviewer’s comment, Fig. 5f and Fig.5h were modified.

d. Methods regarding this experiment: depth of RNA seq? please provide more details on the sequencing procedure.

- The depth of RNA sequencing was 40 million single end reads per sample, as mentioned in the manuscript (Supplementary Materials page 16, line 342-345):

“Library preparation and total RNA-sequencing: Further evaluation of the RNA, RNA library generation, and sequencing were carried out by StarSEQ GmbH (Mainz, Germany). Sequencing was performed on an Illumina NextSeq 500 sequencer with High Output chemistry and minimum output of 40 million single end reads (75 bp) per sample.”

- Additionally, we included four supplementary tables to show in detail the results of the RNA-seq (Supplementary Tables 7-10).

e. DRD1 is also upregulated and is known to affect PL glutamatergic activity – authors could overexpress this gene and observe if it leads to a similar/distinctive phenotype - this would be a nice control experiment to validate DRD2 overexp data

- We agree with the reviewer that it would be interesting to start a new research project to decipher the role of Drd1 gene in mPFC of addicted mice. However, this experiment is out of the scope of the present work. We included a new paragraph in the discussion section to address the possible role of drd1 in the mPFC (page 23, line 517-525)

“On the other hand, Drd1 gene, D₁R in the striatum conforms the direct pathway (GO), which promotes the approach to addiction-like behavior opposite to the D₂R stimulating the indirect pathway (NO GO) (Koob and Volkow 2016). In contrast to the striatum, D₁R stimulation in the mPFC decreases release of glutamate onto L5 pyramidal cells (Gao, Krimer, and Goldman-Rakic 2001) similar to D₂R does (Real et al. 2018). Thus, the increased D₁R mRNA levels in PFC of the addicted mice could also contribute to the decreased excitability of the glutamatergic mPFC neurons projecting to NAc. Therefore,

it could be of interest to explore in future studies whether overexpression of D_2R in mPFC can also modify the development of food addiction in future studies.”

7. Fig. 6:

a. a. is wrong, please check

- We corrected this mistake in the Fig. 6a (currently Fig. 5a). The diagram has now been corrected to reflect properly the *Drd2* overexpression manipulation.

b. Provide evidence of D2R overexpression – qPCR or other technique – this is a major issue, as authors do not present any evidence of DR2 overex. Also, what is the correlation between D2R levels and phenotype (if any)?

- As the reviewer suggests, we performed a quantitative real-time PCR to determine *Drd2* mRNA levels in mPFC of control and in D_2R overexpressing mice (Supplementary Fig.8d). We found a 40-fold increase of *Drd2* gene expression levels in D_2R overexpressing mice as compared to control mice. Correlation between *Drd2* mRNA levels and behavioral outcome is not possible because those mice were employed for electrophysiology studies.

This result is added in the results section as follows (page 16, line 353-355):

“Quantitative real-time PCR showed 40-fold increased mRNA levels of Drd2 in mice overexpressing D₂R as compared to control mice (Supplementary. Fig.8d).”

c. Overexpression of D2R: are the effects mediated by overexpression at dendrites? Or pre-synaptic expression of D2R? at terminals? IHC experiments would show where is the receptor overexpressed (at both sites probably)

- This question was addressed by electrophysiological experiments in the mPFC and in the NAc using quinpirole in mice overexpressing D₂R. Results in the mPFC showed a decrease in the firing rate and membrane resistance with an increase in the current needed to evoke a single action potential (rheobase) in mice overexpressing D₂R compared to mice injected with AAV-control. In the NAc, a reduction in the EPSCs amplitude after the D₂R agonist quinpirole was also obtained in mice overexpressing D₂R, suggesting that the D₂R at glutamatergic PL-NAc projections has an effect in both areas the PL and in the NAc.

Moreover, we performed additional IHC experiments showing overexpressed D₂R in the neuropil of the prelimbic cortical neurons, i.e. at postsynaptic sites in D₂R overexpressing mice, but we could not reveal overexpression of D₂R in NAc of these mice using this particular technique. This effect could be due to a limitation of the IHC approach since we can not detect endogenous D₂R in PL nor in the NAc of control mice under our experimental conditions (supplementary Fig 8 f-i).

The fact that endogenous D₂R levels were not revealed by IHC assays in these conditions, underlies the difficulties to visualize a relevant D₂R expression with these techniques. Our electrophysiological approach was sensitive enough to reveal a response suggesting that the overexpression of D₂R occurred in the postsynaptic dendrites of the PL and in the presynaptic glutamatergic terminals in the NAc although IHC techniques only allowed to confirm the overexpression in the prelimbic.

d. DRD2 overexpression was found in the PFC of addicted animals, a very interesting finding. Then authors overexpress D2R in PL-NAc neurons. Are the behavioural effects due to overexpression of D2R locally in the PL or overexpression in the terminals in the NAc? Having said this, it is very difficult to speak in a “selective top-down cortical pathway”. (please see my comments above)

- We added a new sentence in the results and discussion sections to highlight this issue:

- “Therefore, these electrophysiological results reveal that the overexpression of D2R at glutamatergic PL-NAc projections has an effect in both areas the PL and in the NAc.” (page 17, line 374-376)
- “In particular, upregulation of D2R in postsynaptic dendrites in the PL and in the presynaptic terminals in the NAc diminished the excitability of the pyramidal neurons in the PL projecting to NAc core.” (page 25, line 557-559)

Chemogenetics: Authors always register in L5 (fig. 4, 6), but then assume that the reduced PL activity is observed in less excitation of NAc core. Why did authors did not perform recordings in the NAc core to show exactly this? If one thinks this is the mechanism, then this is crucial.

- As the reviewer suggested, we performed recordings in the NAc core to show that our chemogenetic manipulation decreased the excitability of this area due to the reduced glutamatergic tone coming from the PL (before Fig.4 and current Fig. 3). These results are detailed in Fig.3e-f and in the results section (page 11, line 245-251):

“Importantly, this PL-NAc core projections inhibited the medium spiny neurons (MSNs) of the NAc as shown by the cumulative probability of the amplitude of all signals registered and by the decreased changes of mEPSCs frequency in the NAc after CNO application in mice expressing hM4Di receptors (Fig. 3e-f). No changes in the amplitude nor in the resting membrane potential were reported (Supplementary Fig. 6ef-g). Together these results suggested that the chemogenetic inhibition of the PL-NAc core neurons leads to a reduced glutamatergic tone in the NAc.”

8. Again, in the same line of the previous comment, The hypothesis of the vulnerable phenotype is that the overexpression of D2R in PL terminals is supposed to decrease glutamatergic signalling in NAc core – authors register PL activity, but they do not present

evidence of NAc activity in response to PL stimulation in overex animals. This is crucial to show that indeed this is the “vulnerability” pathway

- Following this suggestion, we measured the glutamatergic transmission in the NAc in D₂R overexpressing mice. These results are detailed in Fig.5e-f and in the results section (page 17, line 368-374):

“Furthermore, we investigated whether this reduced excitability in the PL-NAc core neurons modulates the synaptic glutamatergic transmission in the NAc. Whole-cell recordings in the NAc confirmed a reduction in the changes of mEPSCs frequency accompanied by a sustained difference in the cumulative probability of the amplitude of all the signals registered (Fig. 5 e-f). No changes in the amplitude nor in the resting membrane potential were reported (Supplementary Fig. 9e-f).”

9. Discuss if CB1-KO effects occur mostly at PFC layers or at terminals in the NAc

- We included new electrophysiological experiments to answer this question. Recordings of mEPSCs in the NAc are now included in the revised Fig. 2. We added several paragraphs in the discussion section to explain these new results as follows:
- *“This resilient phenotype was mediated by increased synaptic glutamatergic transmission in mPFC and in NAc. Specifically, our electrophysiological recordings of mEPSCs frequency in Glu-CB1-KO mice revealed an enhanced excitatory synaptic transmission onto pyramidal glutamatergic neurons in L5 of the PL cortex and in their targeted MSNs in the NAc. This effect was specific of glutamatergic transmission, since the synaptic GABAergic transmission in these mutants was not altered in both brain regions.”* (page 19, line 424-430).
- *“Furthermore, an additional experiment of electrophysiology ex vivo in the mPFC and NAc with the application of the CB₁R agonist WIN55,212-2 in mutants and in the WT mice has been performed. With this experiment, we validated at functional level the deletion of CB₁R at glutamatergic presynaptic terminals both in PL and in NAc of Glu-*

CB1-KO. As it was expected, the amplitude of fPSPs in the mPFC and the EPSCs in the NAC were strongly reduced in WT mice after WIN55,212-2 application compared to the baseline, which indicates a functional activation of CB₁R. In the mutants, the fPSPs amplitude was not diminished in the mPFC confirming the absence of the CB₁R at presynaptic glutamatergic terminals. On the other, in the NAC, the EPSCs amplitude was slightly decreased after WIN55,212-2 application, but in a significantly lower extend than in WT mice possibly due to the presence of CB₁R in other non-excitatory cell-types (Marsicano and Kummer 2008; Robbe et al. 2001).” (discussion section, page 20, line 442-452).

-

10. Authors should refer if they performed IHC analysis to detect correct viral expression for all animals used in the experiments. (I assume they did, so this should be referred)

- Yes, we performed immunofluorescence and confocal analysis to detect the correct viral expression for all animals used in the experiments. Accordingly, we highlighted this issue in the manuscript and in the methods section as follows:
- In the results section (page 11, line 233-235):
“mCherry and Cre recombinase were visualized by immunofluorescence to verify the injection site of the AAVs and the retrograde transport of Cre (Fig. 3b).”
- In the results section (page 16, line 347-349):
“We first verified the AAV injection site by immunofluorescence against mVenus and Cre recombinase (Fig. 5b and Supplementary. Fig. 10g-h)”
- In the supplementary results (page 24, line 509-510)
“Monitoring of mCherry expression allowed to verify injection sites (Supplementary Fig. 4b).”
- In the supplementary methods (page 9, lines 191-207):
*“To detect the viral expression in all the experiments we visualized each reporter as it is listed below:
For the detection of AAV8-hSyn-DIO-hM4D(Gi)-mCherry and AAV8-hSyn-DIO-mCherry in Nex-cre mice, we directly visualized mCherry in the confocal microscope. mCherry is a bright red monomeric fluorescent protein that was clearly visible in our experimental conditions without performing an immunofluorescence.*

For the detection of AAV8-hSyn-DIO-hM4D(Gi)-mCherry and AAV8-hSyn-DIO-mCherry in PL, and AAVrg-pmSyn1-EBFP-Cre targeting the NAc, we performed an immunofluorescence immunocytochemistry against Cre to detect the correct viral infection in the NAc and visualized the labelled Cre and the mCherry in the confocal microscope.

For the detection of AAV2/1-hSyn-DIO-SF-D2R(L)-IRES-mVenus and AAV1/2-hSyn-floxstop-hrGFP in the PL and AAVrg-pmSyn1-EBFP-Cre targeting the NAc, we performed an immunofluorescence immunocytochemistry against mVenus/GFP and against Cre to detect the correct viral infection in both brain areas. We visualized the labelled mVenus or GFP, and the labelled Cre in the confocal microscope. In order to clarify the expression of D2R at post- and pre-synaptic terminals, we performed a double immunohistochemistry analysis using anti-D2R and anti-mVenus/GFP antibodies.”

11. Why do authors opt not to include correlation data in the main figures?

- According to the reviewer's suggestion, the correlation data was moved to the main figures (Fig. 3 and Fig. 5).

12. I suggest that the Title should be changed to match the data

- As the reviewer suggested we changed the title to match better with the data.
- **Old title:** A specific top-down cortical pathway controls resilience versus vulnerability to develop food addiction.
- **New title:** A specific prelimbic-nucleus accumbens pathway controls resilience versus vulnerability to develop food addiction.

13. One thing that was not clear for me, but maybe I missed it, was the fact that in experiment of fig. 4 and 6, animals, test groups only present significant changes in the compulsivity score. I would expect changes also in persistence to response. What is the explanation?

- We have now explained this observation. Indeed, persistence to response is a different endophenotype compared to compulsivity. Since the subject has difficulty stopping food use or limiting food intake after a period of extinction learning. This behavior is hypothesized to recruit different brain areas and networks directly involved in habit formation and extinction learning, such as dorsal striatum and hippocampal pathways (Schmitzer-Torbert et al. 2015). To clarify this concept, we introduced the following sentence in the discussion section (page 22, line 485-493):

“The other addiction criteria, persistence to response and motivation, were not modified with this pathway modulation. These criteria represent different endophenotypes from compulsivity and consequently, the neuronal pathways recruited could be different. Persistence to response reflects the difficulty of mice to stop food seeking. This behavior is related to a persistent desire or unsuccessful efforts to cut down the response due to a habit formation or disruption of extinction learning and has been reported to involve dorsal striatal and hippocampal pathways (Schmitzer-Torbert et al. 2015). In turn, motivation is apparently directly related to reward processing and has been associated to VTA-NAc pathways (Lindgren et al. 2017)”

Minor

1. Legend in sup fig2 is wrong. Depicted are WT animals and the “increased transmission” is not observed

- The figure legend in the supplementary Fig. 2 was changed to: *“Blockade of WIN55,212-2 inhibitory effect in mPFC synaptic transmission by selective CB₁R antagonist rimonabant in WT animals”.*

2. Page 6 “Highly suppressed response” please rephrase this sentence, the difference in compulsivity is subtle. In fact, there are just some WT animals that make it go up, otherwise it would be very similar.

- We rephrased this sentence as follows (page 6, lines 123-124):
“Indeed, a suppressed response in front of negative consequences was revealed in Glu-CB1-KO mice compared to WT mice (P<0.05, Supplementary Fig. 1c).”

3. Page 6 last sentence is very confusing

- We changed this sentence as it is detailed below (page 7, lines 143-146):
“Thus, the deletion of the CB₁R in dorsal telencephalic glutamatergic neurons is a protective factor for preventing the development of food addiction but, as expected in a multifactorial disease, the mutation of one single gene is not enough to totally stop the addiction process.”

4. Consider using other colour scheme throughout the manuscript– for example in sup fig 1f-h it is impossible to distinguish between black and dark blue. Red dots in red bars are also very difficult to see – e.g. Fig 4g

- We changed the resolution of all these figures for a better distinction between groups and individual points.
- 5. Page 9 first sentence confusing, too many claims**
- We modified this sentence as follows (page 10, lines 216-219):
“Based on the increased excitatory transmission in mPFC of Glu-CB1-KO mice resilient to addiction, we hypothesized that hypoactivity of glutamatergic transmission in mPFC would promote addictive-like behavior in WT mice when exposed to the palatable food addiction model.”
- 6. Page 9 last sentence is confusing**
- We changed the sentence as it is specified below (page 12, lines 252-262):
“Using this approach, we expected that selective inhibition of the PL-NAc core projections would decrease the inhibitory control during food operant training leading a more susceptible phenotype to develop addiction-like behavior. Therefore, we used the early food addiction protocol (Fig. 3g) to train WT mice (n=32) to self-administer chocolate-flavored pellets in the operant chambers under FR1 (2 sessions) and FR5 (3 sessions) schedule of reinforcement before AAVs injection and under FR5 (4 sessions) after injection to recover the basal levels of responding.”
- 7. Page 11 – refer that recordings are made in L5**
- According to this suggestion, we have specified that recordings were made in L5 (page 11, lines 235-238).
“Whole-cell current clamp recordings performed in PL L5 of visually identified hM4Di-mCherry expressing neurons confirmed that CNO activation of hM4Di receptor inhibited the activity of PL-NAc core pyramidal neurons.”
- 8. Legend fig 4. Remove “Loss of inhibitory control”, as you did not directly evaluate this parameter**
- We removed this part of the sentence.
- 9. Page 13: “Finally, a positive correlation between the number of criteria reached and the values of each addiction criterion was found in both groups (Supplementary Fig. 4i-n).” – rephrase, as in saline group the correlations are not significant for two of the tests. (the same comment for page 17)**

- We modified the text accordingly as follows (page 13, line 280-284):
“Finally, a positive correlation between the number of criteria reached and the values of each addiction criterion was found in CNO group in the three criteria and in saline group only in the criterion of compulsivity (Fig. 3m-o), and the classification of addicted mice showed higher values in both saline and CNO treated mice (Supplementary Fig. 6k-m).”

10. Fig5f a should be *

- We modified the symbol a to ★ .

11. page 14 -“In fact, Fos mRNA levels correlate with the increased number of reinforcers in WT as compared with Glu-CB1-KO immediately prior to sample collection (Fig. 5d).”- this correlation is not shown anywhere, remove sentence

- This sentence was removed in accordance with the reviewer suggestion.

12. page 15: “in mPFC related to addiction” – food addiction

- The sentence was modified as requested: *“in mPFC related to food addiction”*.

13. Fig 6b: there is substantial background in the slice; authors should also provide better quality and magnified figures (in sup.)

- The images have been modified to provide better quality. In the supplementary Fig. 10g-h we added magnified pictures of the labelled cells.

14. page 15: “we confirmed low endogenous Drd2 mRNA expression in PL by in situ hybridization (ISH) (Supplementary Fig. 6a).” – compared to Cpu and NAc, I assume? No quantification is provided. A is out of place in the figure a.

- We rephrased the sentence to *“First, we confirmed under basal conditions low endogenous Drd2 mRNA expression in PL as compared to NAc and caudate putamen by in situ hybridization (ISH)”* (page 16, line 341-343). The differences in expression

between PL and CPu / NAc are clearly apparent as seen from the ISH experiments. Therefore, we consider that detailed quantification does not provide additional mechanistic insights into our work. The mistake showing a letter in the middle of the panel “a” has been corrected.

15. page 15: “Cre recombinase mRNA is present in PL, and that importantly this mRNA is coexpressed with the endogenous Drd2 mRNA” -. Provide quantification (in sup is ok)

- In agreement with this comment, a quantification of co-expressed mRNAs is provided in the new version of the manuscript (please, see Supplementary Results):

“Quantification of cells co-expressing Drd2 and Cre mRNA

For quantification of cells co-expressing Drd2 and Cre mRNA in PL, we counted about 500 cells each on sections from AAV-retrograde-Cre (AAVrg-pmSyn-EBFP-Cre) injected animals. Sections were hybridized with riboprobes for Drd2 and Cre. There were less Cre-positive than Drd2-positive cells, showing that the injected Cre-expressing virus into NAc targeted the region of interest, i.e., PL, but not all Drd2 expressing cells in PL.” (page 26, line 554-558)

16. no effect of quinpirole in activity of L5 neurons of control animals – this is surprising, discuss (sup fig 7)

- The lack of effect after quinpirole application in control animals could be explained by the very low levels of endogenous D2R in mPFC according to RNA-seq data. In agreement, our ISH experiments showed that there is endogenous Drd2 mRNA expression in this type of cells (L5) but at very low levels close to the limit of detection. Further, our approach consisted in using the minimum optimal quinpirole concentration to reveal an effect in mice overexpressing D2R and we could speculate that a higher concentration of the D2R agonist would be needed to see a stronger effect in both groups.

17. page 16: remove “presumably by endogenous dopamine.”

- According to the reviewer the sentence “presumably by endogenous dopamine.” has been removed.

18. What is the expression of D2R in CB1KO animals?

- Total CB1-KO mice showed a reduced response during operant conditioning maintained by chocolate pellets (Mancino et al. 2015). Therefore, it would be interesting to analyze the expression of D2R in these mice. However, it is well known that compensatory processes are usually associated with the global and life-long deletion of a particular gene. For this reason, current neurogenetic experiments in mouse prefer conditional, i.e., cell type and/or region-specific gene modifications. Accordingly, we provide *Drd2* gene expression analyses in Glu-CB1-KO mice using RNA-seq and qPCR experiments (Fig. 4e).

Methods

19. name ethical committee authorization number

- We added the required information in the methods section as it is specified below (page 2, line 32-36):

“All experimental protocols were performed in accordance with the guidelines of the European Communities Council Directive 2010/63/EU and approved by the local ethical committee (Comitè Ètic d'Experimentació Animal-Parc de Recerca Biomèdica de Barcelona, CEEA-PRBB, agreement N°9687).”

20. Stats: Did authors check for outliers? How did authors perform chi-square analyses – refer in methods

- We did not checked for outliers. In our model, we are interested in extreme behavioral phenotypes, i.e, vulnerable and resistant mice. The chi-square analyses were performed using the IBM SPSS statistics software to compare the percentage of addicted mice with the non-addicted ones. For this purpose, we compared the observed frequencies with the frequencies obtained in the control group.
- According to the reviewer’s comment, these details are specified in the statistical analysis section from methods (page 22, line 475-477).

“The chi-square analyses were performed to compare the percentage of addicted mice with the non-addicted ones, we compared the observed frequencies with the frequencies obtained in the control group.”

21. Page 14: how was the mPFC removed? Macrodissection? please refer

- The mPFC was removed by microdissection as now referred in the methods sections (page 16, line 333-335).

“The mPFC was isolated by microdissection according to the following coordinate from Paxinos and Franklin 2001 40 (AP +1.98 mm) and the samples were placed in individual tubes, frozen on dry ice, and stored at –80°C until RNA isolation for the RNA-sequencing.”

22. Provide more info about PCA analysis

- We added this information in the Materials and Methods section (page 17, line 357-260).

“Variability between the addicted and non-addicted mice was determined using PCA analysis from “plotPCA” function of DEseq package. For PCA analysis, top varying 500 genes were selected.”

23. DELTA DELTA CT method was used for qPCR but is unclear what was the control group? Which housekeeping genes were used for qPCR? Actin? Usp? More than one? How was the normalization done? for one of the genes or both? Please provide more information.

- In qPCR validation and RNA-seq analysis, the same control groups were used, i.e., non-addicted and WT mice were both used as the control groups in Fig. 4f and h, respectively.
- We added the following paragraph in the supplementary materials to clarify the use of housekeeping genes (page 18, line 376-382)
- *“We measured the gene expression of three different house keeping genes (Tbp, Usp11, actin) in these samples using qPCR in order to verify that the expression of these genes was not affected by the operant model of food addiction used, as represented in Supplementary Material (Supplementary Fig. 7c-e). Regarding the normalization of differentially expressed genes in qPCR validation, all validated differentially expressed genes were normalized using Tbp as housekeeping gene, although the same significant changes were also found using the other two housekeeping genes (Usp11, actin).”*

24. Provide serotypes for all virus used

- In this study, we used 5 different viruses with specific serotypes. They are mentioned in “viral vectors” of the methods section (page 8):

- AAV-hM4Di-DREADD → AAV8-hSyn-DIO-hM4D(Gi)-mCherry
- AAV-control-DREADD → AAV8-hSyn-DIO-mCherry
- AAV-retrograde-Cre → AAVrg pmSyn1-EBFP-Cre. This viral vector preparations were produced with the rAAV2-retro helper plasmid.
- AAV-D2R → AAV2/1-hSyn-DIO-SF-D2R(L)-IRES-mVenus
- AAV-control-D2R → AAV1/2-hSyn-floxstop-hrGFP

Sup material

25. sup fig 3b-e – sham operated mice? The best adequate control would be the AAV control DREADD – authors should explain the reason to use sham operated animals.

- We agree with the reviewer, the adequate control for this experiment is the injection of the AAV-control-DREADD. This was the control used in our study as now specified in the methods section (it was expressed incorrectly in the supplementary fig. 3b-e):
- *“b-e, Lack of CNO-induced effects in mice injected with AAV-control-DREADD either treated with saline (n=5) or CNO (n=5).”*

26. Sup table 4 legend typo

- Corrected.

27. Page 13: “Two overlapping clusters were observed, allowing a general separation of the addicted from non-addicted mice” – sup. Fig 5b – this is not clear to me, and no stats is provided to support this claim.

- We corrected this sentence and supplementary fig. 5b. What we liked to illustrate with Supplementary Fig. 5b (currently supplementary Fig.7b) is that plotting the first two principal components of this multidimensional dataset allows to visualize a considerable separation between addicted and non-addicted subtypes with respect to their gene expression profiles. This separation, however, is not strict since two “non-addicted” animals showed a transcriptome more similar to the addicted mice, whereas the profiles of three addicted individuals were similar to the group of the non-addicted animals. This grouping of the animals was meant with the term “overlapping cluster”. We elaborated the description of the PCA-plot in the manuscript to clarify the meaning of Supplementary. Fig. 5b. Furthermore, we added the percentage of variance captured by the first two principal components.
- We have changed the phrase for clarity as follows (page 14, line 300-301):

“Two different clusters were observed for the addicted and non-addicted mice (Supplementary Fig. 7b).”

28. Sup fig 5b: ok to separate between addicted and non-addicted but color code WT and CB1 animals in the PCA analysis

- In the revised manuscript, the figure now contains the required information.

29. Sup fig 7 legend is incomplete, not mentioned what data are from controls vs DR2 overexp.

- The figure legend was improved to clarify the data from controls vs D2R overexpression.

30. Sup fig 9: in the PL terminals in the NAc, authors present CB1 on the resilient phenotype but no D2R and D2R in vulnerable but no CB1 – this is not accurate as CB1 is expressed also in terminals of vulnerable mice right? (and so on)

- As the reviewer suggests, we now present in the resilient phenotype the D2R and in the vulnerable phenotype the CB₁R (Supplementary Fig. 11a-b)

31. Sup. Fig 9: typo in dopaminergic neuron (appears neuron)

- Corrected.

Additional Suggestions:

1. Manuscript should be read by an English-native speaker. Some sentences are awkward, some grammatical errors.

2. Consistently authors drive hypothesis based on very sparse data (eg. Last sentence of page 9), I would refrain of doing this. Be more to the point, less inferences, the data speaks per se.

- According to this comment, we rephrased the sentence as follows (page 12, lines 252-255):

“Using this approach, we expected that selective inhibition of the PL-NAc core projections would decrease the inhibitory control during food operant training, leading to susceptibility to develop addiction-like behavior. Therefore, we performed the early food addiction protocol (Fig. 3g).”

3. I usually do not interfere much with discussion – as it is a space for authors to have “more freedom” in their interpretations, but I would refrain from allusions to “therapeutic research possibilities” – page 22. I would “tone down” the conclusion section.

- We modified the conclusion section as suggested by the reviewer (page 26, lines 569-573).

“Our results provided new mechanistic underpinnings of food addiction, not only valued for the field of food addiction, but also for other psychiatric disorders with alterations in compulsive behavior due to the transdiagnostic nature of this concept. In addition, our findings pave the way for prevention measures, by identifying mechanisms required for strengthening the resilient phenotype.”

References

- Van Aerde, Karlijn I. and Dirk Feldmeyer. 2015. “Morphological and Physiological Characterization of Pyramidal Neuron Subtypes in Rat Medial Prefrontal Cortex.” *Cerebral Cortex* 25(3):788–805.
- Bellocchio, Luigi, Pauline Lafenêtre, Astrid Cannich, Daniela Cota, Nagore Puente, Pedro Grandes, Francis Chaouloff, Pier Vincenzo Piazza, and Giovanni Marsicano. 2010. “Bimodal Control of Stimulated Food Intake by the Endocannabinoid System.” *Nature Neuroscience* 13(3):281–83.
- Blum, K., E. R. Braverman, R. C. Wood, J. Gill, C. Li, T. J. Chen, M. Taub, A. R. Montgomery, P. J. Sheridan, and J. G. Cull. 1996. “Increased Prevalence of the Taq I A1 Allele of the Dopamine Receptor Gene (DRD2) in Obesity with Comorbid Substance Use Disorder: A Preliminary Report.” *Pharmacogenetics* 6(4):297–305.
- Gao, W. J., L. S. Krimer, and P. S. Goldman-Rakic. 2001. “Presynaptic Regulation of Recurrent Excitation by D1 Receptors in Prefrontal Circuits.” *Proceedings of the National Academy of Sciences of the United States of America* 98(1):295–300.
- Gearhardt, Ashley N., William R. Corbin, and Kelly D. Brownell. 2016. “Development of the Yale Food Addiction Scale Version 2.0.” *Psychology of Addictive Behaviors* 30(1):113–21.
- Gordon, Eliza, Aviva Ariel-Donges, Viviana Bauman, Lisa Merlo, Eliza L. Gordon, Aviva H. Ariel-Donges, Viviana Bauman, and Lisa J. Merlo. 2018. “What Is the Evidence for ‘Food Addiction?’ A Systematic Review.” *Nutrients* 10(4):477.

- Häring, Martin, Nadine Kaiser, Krisztina Monory, and Beat Lutz. 2011. "Circuit Specific Functions of Cannabinoid CB1 Receptor in the Balance of Investigatory Drive and Exploration." edited by H. A. Burgess. *PLoS One* 6(11):e26617.
- Heidbreder, Christian A. and Henk J. Groenewegen. 2003. "The Medial Prefrontal Cortex in the Rat: Evidence for a Dorso-Ventral Distinction Based upon Functional and Anatomical Characteristics." *Neuroscience & Biobehavioral Reviews* 27(6):555–79.
- Kneussel, Matthias and Wolfgang Wagner. 2013. "Myosin Motors at Neuronal Synapses: Drivers of Membrane Transport and Actin Dynamics." *Nature Reviews Neuroscience* 14(4):233–47.
- Koob, George F. and Nora D. Volkow. 2016. "Neurobiology of Addiction: A Neurocircuitry Analysis." *The Lancet Psychiatry* 3(8):760–73.
- Lindgren, Elsa, Kyle Gray, Gregg Miller, Ryan Tyler, Corinde E. Wiers, Nora D. Volkow, and Gene-Jack Wang. 2017. "Food Addiction: A Common Neurobiological Mechanism with Drug Abuse." *Frontiers in Bioscience (Landmark Edition)* 23:811–36.
- Lutz, Beat, Giovanni Marsicano, Rafael Maldonado, and Cecilia J. Hillard. 2015. "The Endocannabinoid System in Guarding against Fear, Anxiety and Stress." *Nature Reviews Neuroscience* 16(12):705–18.
- Mancino, S., A. Burokas, J. Gutierrez-Cuesta, M. Gutierrez-Martos, E. Martin-Garcia, M. Pucci, A. Falconi, C. D'Addario, M. Maccarrone, and R. Maldonado. 2015. "Epigenetic and Proteomic Expression Changes Promoted by Eating Addictive-Like Behavior." *Neuropsychopharmacology*. 10.
- Marsicano, Giovanni and Rohini Kunner. 2008. "Chapter 10. Anatomical Distribution of Receptors, Ligands and Enzymes in the Brain and in the Spinal Cord: Circuitries and Neurochemistry." Pp. 161–201 in *Cannabinoids and the brain*, edited by K. A. (ed). New York: Springer.
- Moorman, David E., Morgan H. James, Ellen M. McGlinchey, and Gary Aston-Jones. 2015. "Differential Roles of Medial Prefrontal Subregions in the Regulation of Drug Seeking." *Brain Research* 1628:130–46.
- Nestler, Eric J. 2013. "Cellular Basis of Memory for Addiction." *Dialogues in Clinical Neuroscience* 15(4):431–43.
- Quintana, Albert, Elisenda Sanz, Wengang Wang, Granville P. Storey, Ali D. Güler, Matthew J.

- Wanat, Bryan A. Roller, Anna La Torre, Paul S. Amieux, G. Stanley McKnight, Nigel S. Bamford, and Richard D. Palmiter. 2012. "Lack of GPR88 Enhances Medium Spiny Neuron Activity and Alters Motor- and Cue-Dependent Behaviors." *Nature Neuroscience* 15(11):1547–55.
- Real, Joana I., Ana Patrícia Simões, Rodrigo A. Cunha, Samira G. Ferreira, and Daniel Rial. 2018. "Adenosine A_{2A} Receptors Modulate the Dopamine D₂ Receptor-Mediated Inhibition of Synaptic Transmission in the Mouse Prefrontal Cortex." *European Journal of Neuroscience* 47(January):1127–34.
- Robbe, D., G. Alonso, F. Duchamp, J. Bockaert, and O. J. Manzoni. 2001. "Localization and Mechanisms of Action of Cannabinoid Receptors at the Glutamatergic Synapses of the Mouse Nucleus Accumbens." *The Journal of Neuroscience : The Official Journal of the Society for Neuroscience* 21(1):109–16.
- Schmitzer-Torbert, Neil, Steven Apostolidis, Romeo Amoa, Connor O’Rear, Michael Kaster, Josh Stowers, and Robert Ritz. 2015. "Post-Training Cocaine Administration Facilitates Habit Learning and Requires the Infralimbic Cortex and Dorsolateral Striatum." *Neurobiology of Learning and Memory* 118:105–12.
- Tan, Zhu Jun, Yun Peng, He Ling Song, Jing Jing Zheng, and Xiang Yu. 2010. "N-Cadherin-Dependent Neuron-Neuron Interaction Is Required for the Maintenance of Activity-Induced Dendrite Growth." *Proceedings of the National Academy of Sciences of the United States of America* 107(21):9873–78.
- Urade, Yoshihiro and Osamu Hayaishi. 2000. "Biochemical, Structural, Genetic, Physiological, and Pathophysiological Features of Lipocalin-Type Prostaglandin D Synthase." *Biochimica et Biophysica Acta - Protein Structure and Molecular Enzymology* 1482(1–2):259–71.
- Vertes, Robert P. 2006. "Interactions among the Medial Prefrontal Cortex, Hippocampus and Midline Thalamus in Emotional and Cognitive Processing in the Rat." *Neuroscience* 142(1):1–20.
- Volkow, Nora D., Joanna S. Fowler, Gene-Jack Wang, Robert Hitzemann, Jean Logan, David J. Schlyer, Stephen L. Dewey, and Alfred P. Wolf. 1993. "Decreased Dopamine D2 Receptor Availability Is Associated with Reduced Frontal Metabolism in Cocaine Abusers." *Synapse* 14(2):169–77.
- Wang, G. J., N. D. Volkow, J. Logan, N. R. Pappas, C. T. Wong, W. Zhu, N. Netusil, and J. S.

Fowler. 2001. "Brain Dopamine and Obesity." *Lancet (London, England)* 357(9253):354–57.

Wang, Wengang, Dennis Dever, Janet Lowe, Granville P. Storey, Anita Bhansali, Emily K. Eck, Ioana Nitulescu, Jessica Weimer, and Nigel S. Bamford. 2012. "Regulation of Prefrontal Excitatory Neurotransmission by Dopamine in the Nucleus Accumbens Core." *Journal of Physiology* 590(16):3743–69.

REVIEWERS' COMMENTS:

Reviewer #1 (Remarks to the Author):

Authors have made considerable effort in addressing the comments thoroughly and the revised manuscript has improved in clarity and rigor, strengthening its conclusions and therefore its implications.

Additional experiments were performed. Recordings from NAc MSNs in Figure 2 further validate the genetic mouse model. There is also new data on inhibitory transmission, which adds to the selectivity of the perturbation. This is great addition. My original comment extended a bit further but by re-reading it I realized that it was not very clear. I suggested testing whether the CB1R modulation of GABA transmission was intact, which is not really shown but it is not required.

I also consider that the manuscript is improved by the additional consideration of body weight as a factor for the behavioral changes, which is now evaluated in Suppl. Figure 2. And the additional recording from NAc MSNs as requested by Reviewer #3 and now added in Figure 2 and 4 also improve the paper.

I also agree with Reviewer #2 on the suggestion to revised the term "addicted" and "addiction" for "addictive-like", which the authors only partially did and in my opinion will also improve if it was done across the manuscript.

Reviewer #2 (Remarks to the Author):

The authors have satisfactorily addressed my concerns.

Daniele Piomelli

Reviewer #3 (Remarks to the Author):

Comments

Authors performed additional experiments that strengthen the manuscript and answered almost all of my concerns.

Below there are some minor points that still need clarification.

- Comment 2- "We cannot discard the possible involvement of the anterior cingulate cortex in this behavior, since we also observe residual viral expression in this brain region".
- Ephys for NAc – why did authors decided to present data as "cumulative probability of the amplitude of all the signals registered"? please include typical ephys data in addition to this.
- Fig4f: * is de-formatted
- Line 144-146 "protective factor"? – this sentence should be toned down.
- Line 207-209- not clear to me- what is your explanation for the partial effect?, and the reference to the statement is missing (line 209).
- Line 349- sup fig 10g and h is wrong.
- Line 557-559 – confusing, maybe just say " In particular, upregulation of D2R (either in postsynaptic dendrites in the PL and or in presynaptic terminals in the NAc) diminished the excitability of the pyramidal neurons in the PL projecting to NAc core.
- Line 375-376 grammatically wrong
- Comment 10: maybe I did not explained myself well, but my idea was that authors would show that the viral approach worked – PL-NAc pathway. Authors performed also cre IF, which is one more confirmation, so: well done!

- Title: change controls to modulates. It is more accurate.
- Comment 16- this explanation should be mentioned in the manuscript
- Comment 21: it is microdissection because you did nit perform laser microdissection.
- Pl-amy – I understand the time it would take to perform the experiments. But you should discuss that additional studies are needed to understand if this is specific to PL-NAc core or to PL- other brain regions.

Reviewers' comments:

Reviewer #1 (Remarks to the Author):

Authors have made considerable effort in addressing the comments thoroughly and the revised manuscript has improved in clarity and rigor, strengthening its conclusions and therefore its implications.

Additional experiments were performed. Recordings from NAc MSNs in Figure 2 further validate the genetic mouse model. There is also new data on inhibitory transmission, which adds to the selectivity of the perturbation. This is great addition. My original comment extended a bit further but by re-reading it I realized that it was not very clear. I suggested testing whether the CB1R modulation of GABA transmission was intact, which is not really shown but it is not required

I also consider that the manuscript is improved by the additional consideration of body weight as a factor for the behavioral changes, which is now evaluated in Suppl. Figure 2. And the additional recording from NAc MSNs as requested by Reviewer #3 and now added in Figure 2 and 4 also improve the paper.

I also agree with Reviewer #2 on the suggestion to revised the term “addicted” and “addiction” for “addictive-like”, which the authors only partially did and in my opinion will also improve if it was done across the manuscript.

We thank the reviewer for her/his thoughtful comments that helped to improve our article. Accordingly, we corrected the term “addicted” and “addiction” for “addiction-like”.

Reviewer #2 (Remarks to the Author):

The authors have satisfactorily addressed my concerns.

We thank the reviewer for his positive feedback.

Reviewer #3 (Remarks to the Author):

- **Authors performed additional experiments that strengthen the manuscript and answered almost all of my concerns.**

- We thank the reviewer for his/her positive reception of our work that helped strengthening our conclusions. We have included the changes suggested which are specified below point by point.

Below there are some minor points that still need clarification.

- 1. Comment 2- “We cannot discard the possible involvement of the anterior cingulate cortex in this behavior, since we also observe residual viral expression in this brain region”.**
 - We modified the sentence (page 21).
- 2. Ephys for NAc – why did authors decided to present data as “cumulative probability of the amplitude of all the signals registered”? please include typical ephys data in addition to this.**
 - Following the suggestion of the reviewer, we included typical ephys data in the corresponding Figures 3e and 5e.
- 3. Fig4f: * is de-formatted**
 - The ★ of the Fig.4f have been formatted correctly.
- 4. Line 144-146 “protective factor”? – this sentence should be toned down.**
 - Accordingly with the reviewer we toned down the sentence as follows:
 - *“Thus, the deletion of the CB₁R in dorsal telencephalic glutamatergic neurons **could represent a protective factor to food addiction-like behavior** but, as expected in a multifactorial disease, the mutation of one single gene is not enough to totally stop the addiction process.”*
- 5. Line 207-209- not clear to me- what is your explanation for the partial effect?, and the reference to the statement is missing (line 209).**
 - We changed the sentence and added the references to the statement as follows:
 - “This partial effect of WIN55,212-2 in the mutants could be explained by the expression of CB1R in glutamatergic projecting neurons from other areas to the NAc¹⁻³.”*
- 6. Line 349- sup fig 10g and h is wrong.**
 - The sentence is corrected.
- 7. Line 557-559 – confusing, maybe just say “ In particular, upregulation of D2R (either in postsynaptic dendrites in the PL and or in presynaptic terminals in the NAc) diminished the excitability of the pyramidal neurons in the PL projecting to NAc core.**
 - Sentence in page 25 is modified accordingly to the reviewer’s comment.

8. Line 375-376 grammatically wrong

- This sentence is corrected as follows:
- *“Therefore, these electrophysiological results reveal that the overexpression of D₂R at glutamatergic PL neurons projecting to the NAc has an effect in both areas, the PL and NAc.”*

9. Comment 10: maybe I did not explained myself well, but my idea was that authors would show that the viral approach worked – PL-NAc pathway. Authors performed also cre IF, which is one more confirmation, so: well done!

- We thank the reviewer for her/his positive comment.

10. Title: change controls to modulates. It is more accurate.

- The original title has been maintained since we consider that is accurate and provides a direct and clear statement about the main findings of the work.
“A specific prelimbic-nucleus accumbens pathway controls resilience versus vulnerability to food addiction.”

11. Comment 16- this explanation should be mentioned in the manuscript

- This statement was added to the final manuscript in the results section (page 17):
“The lack of effect after quinpirole application in control animals could be explained by the low levels of endogenous D₂R in mPFC according to RNA-seq data. In agreement, our ISH experiments showed endogenous Drd2 mRNA expression in this type of cells (L5), but at low levels close to the limit of detection.”

12. Comment 21: it is macrodissection because you did not perform laser microdissection.

- Accordingly, we modified microdissection for macrodissection.

13. Pl-amy – I understand the time it would take to perform the experiments. But you should discuss that additional studies are needed to understand if this is specific to PL-NAc core or to PL- other brain regions.

- Following the suggestion of the reviewer, we included the following sentence in the discussion section:
“Future studies should be performed to elucidate the role of other pathways from the PL to other downstream areas, such as the amygdala, in the development of food addiction-like behavior.”

References

1. Marsicano, G. & Kuner, R. Chapter 10. Anatomical distribution of receptors, ligands and enzymes in the brain and in the spinal cord: circuitries and neurochemistry. in *Cannabinoids and the brain* (ed. (ed), K. A.) 161–201 (Springer, 2008).
2. Scofield, M. D. *et al.* The nucleus accumbens: Mechanisms of addiction across drug classes reflect the importance of glutamate homeostasis. *Pharmacol. Rev.* **68**, 816–871 (2016).
3. Stuber, G. D. *et al.* Excitatory transmission from the amygdala to nucleus accumbens facilitates reward seeking. *Nature* **475**, 377–382 (2011).